# Evidence for the early emergence of piperaquine-resistant *Plasmodium falciparum* malaria and modeling strategies to mitigate resistance

**Jennifer L. Small-Saunders**[1], **Laura M. Hagenah**[2], **Kathryn J. Wicht**[2], **Satish K. Dhingra**[2], **Ioanna Deni**[2], **Jonathan Kim**[3], **Jeremie Vendome**[4], **Eva Gil-Iturbe**[5], **Paul D. Roepe**[6,7], **Monica Mehta**[1], **Filippo Mancia**[3], **Matthias Quick**[5,8,9], **Margaret J. Eppstein**[10,11,12], **David A. Fidock**[1,2]*

1 Center for Malaria Therapeutics and Antimicrobial Resistance, Division of Infectious Diseases, Department of Medicine, Columbia University Irving Medical Center, New York, New York, United States of America, 2 Department of Microbiology and Immunology, Columbia University Irving Medical Center, New York, New York, United States of America, 3 Department of Physiology and Cellular Biophysics, Columbia University Irving Medical Center, New York, New York United States of America, 4 Schrödinger, Inc., New York, New York, United States of America, 5 Department of Psychiatry, Columbia University Irving Medical Center, New York, New York, United States of America, 6 Department of Chemistry, Georgetown University, Washington, DC, United States of America, 7 Department of Biochemistry and Cellular and Molecular Biology, Georgetown University, Washington, DC, United States of America, 8 Division of Molecular Therapeutics, New York State Psychiatric Institute, New York, New York, United States of America, 9 Center for Molecular Recognition, Columbia University Irving Medical Center, New York, New York, United States of America, 10 Vermont Complex Systems Center, University of Vermont, Burlington, Vermont, United States of America, 11 Department of Computer Science, University of Vermont, Burlington, Vermont, United States of America, 12 Translational Global Infectious Diseases Research Center, University of Vermont, Burlington, Vermont, United States of America

* df2260@cumc.columbia.edu

**Data Availability Statement:** All relevant data are within the manuscript and its Supporting Information files.

## Abstract

Multidrug-resistant *Plasmodium falciparum* parasites have emerged in Cambodia and neighboring countries in Southeast Asia, compromising the efficacy of first-line antimalarial combinations. Dihydroartemisinin + piperaquine (PPQ) treatment failure rates have risen to as high as 50% in some areas in this region. For PPQ, resistance is driven primarily by a series of mutant alleles of the *P. falciparum* chloroquine resistance transporter (PfCRT). PPQ resistance was reported in China three decades earlier, but the molecular driver remained unknown. Herein, we identify a PPQ-resistant *pfcrt* allele (China C) from Yunnan Province, China, whose genotypic lineage is distinct from the PPQ-resistant *pfcrt* alleles currently observed in Cambodia. Combining gene editing and competitive growth assays, we report that PfCRT China C confers moderate PPQ resistance while re-sensitizing parasites to chloroquine (CQ) and incurring a fitness cost that manifests as a reduced rate of parasite growth. PPQ transport assays using purified PfCRT isoforms, combined with molecular dynamics simulations, highlight differences in drug transport kinetics and in this transporter's central cavity conformation between China C and the current Southeast Asian PPQ-resistant isoforms. We also report a novel computational model that incorporates empirically determined fitness landscapes at varying drug concentrations, combined with antimalarial

**Funding:** JSS is a recipient of a Doris Duke Charitable Foundation Physician Scientist Fellowship and a NIH K08 award (K08 AI163497) and was also supported by the Columbia Integrated Program in Infectious Diseases Research (T32 AI100852; PIs Drs. Anne-Catrin Uhlemann and Magdalena Sobieszczyk). LMH acknowledges support from the Columbia University Graduate Program in Microbiology and Immunology (T32 AI106711; PI Dr. David Fidock) and the NIH (F31 AI15740; PI Laura Hagenah). MJE acknowledges support from the Translational Global Infectious Diseases Research Center (NIH P20 GM125498; PI Dr. Beth Kirkpatrick). We also gratefully acknowledge funding support from the NIH (R37 AI050234 and R01 AI124678 to DAF; R01 AI147628 to FM, DAF, MQ; R01 AI056312 to PDR; and R21 AI159558 to SKD). The funders had no role in study design, data collection and analysis, decision to publish, or preparation of the manuscript.

**Competing interests:** The authors have declared that no competing interests exist.

susceptibility profiles, mutation rates, and drug pharmacokinetics. Our simulations with PPQ-resistant or -sensitive parasite lines predict that a three-day regimen of PPQ combined with CQ can effectively clear infections and prevent the evolution of PfCRT variants. This work suggests that including CQ in combination therapies could be effective in suppressing the evolution of PfCRT-mediated multidrug resistance in regions where PPQ has lost efficacy.

## Author summary

The recent emergence of *Plasmodium falciparum* parasite resistance to the antimalarial drug piperaquine (PPQ) has contributed to frequent treatment failures across Southeast Asia, originating in Cambodia. Here, we show that earlier reports of PPQ resistance in Yunnan Province, China could be explained by the unique China C variant of the *P. falciparum* chloroquine resistance transporter PfCRT. Gene-edited parasites show a loss of fitness and parasite resensitization to the chemically related former first-line antimalarial chloroquine, while acquiring PPQ resistance via drug efflux. Molecular features of drug resistance were examined using biochemical assays to measure mutant PfCRT-mediated drug transport and molecular dynamics simulations with the recently solved PfCRT structure to assess changes in the central drug-binding cavity. We also describe a new computational model that incorporates parasite mutation rates, fitness costs, antimalarial susceptibilities, and drug pharmacological profiles to predict how infections with parasite strains expressing distinct PfCRT variants can evolve and be selected in response to different drug pressures and regimens. Simulations predict that a three-day regimen of PPQ plus chloroquine would be fully effective at preventing recrudescence of drug-resistant infections.

## Introduction

The evolution of drug-resistant *Plasmodium falciparum* asexual blood stage parasites continues to threaten global malaria treatment and control. In 2020, malaria resulted in an estimated 241 million cases and 627,000 deaths [1]. Resistance to the former first line antimalarial, chloroquine (CQ), first emerged in Southeast Asia (SE Asia) and then later swept across Africa, causing substantial increases in mortality [2,3]. CQ was ultimately replaced by the current first-line artemisinin (ART)-based combination therapies (ACTs), composed of a fast-acting ART derivative with a longer-lasting partner drug. ACTs, along with vector control strategies, have substantially reduced the global malaria burden since 2000. However, delayed parasite clearance after ART treatment is now present throughout SE Asia. Resistance has also emerged to the ACT partner drug piperaquine (PPQ), leading to up to 50% treatment failures with this combination in some areas of the Greater Mekong Subregion [4,5]. Mutations in the transmembrane protein *Plasmodium falciparum* chloroquine resistance transporter (PfCRT) have been shown to be the major drivers of high-grade CQ and more recently, PPQ resistance [6–10]. Genome-wide association studies with SE Asian parasites have also associated novel PfCRT mutations with increased *in vitro* survival of PPQ-treated parasites and an increased risk of clinical PPQ treatment failure [11–14]. As new PPQ resistance mutations appear across SE Asia, it is imperative to understand their evolution, how they affect drug susceptibilities

and parasite fitness, how these mutations alter the drug transporter at the molecular level, and whether these evolutionary observations can be harnessed to optimize treatment.

Resistance to CQ evolved independently in SE Asia, the Western Pacific, and South America via different combinations of PfCRT mutations, with the K76T mutation being necessary and ubiquitous across regions. These haplotypes include Dd2 and GB4, present in SE Asia, which differ from the CQ-sensitive wild-type (WT) 3D7 haplotype by eight and six amino acids, respectively [15]. GB4 has now become the predominant mutant PfCRT haplotype in Africa [16,17]. PfCRT mutations have also been implicated in resistance to amodiaquine [18,19], have been demonstrated to impact parasite susceptibility to additional first-line antimalarials [20], and often reduce parasite fitness in the absence of drug pressure [17,21,22]. Fitness in this sense refers to rates of asexual blood stage parasite growth *in vitro*. The interplay between resistance levels, drug pressure, and fitness is evidenced by the resurgence of the more fit, CQ-sensitive parasites in areas of Africa and Asia where CQ has been replaced [23–27].

In Cambodia, the epicenter of PPQ resistance, novel mutations in PfCRT increased in prevalence from <10% in 2011 to >90% by 2016 [9]. These mutations all evolved from the CQ-resistant PfCRT Dd2 haplotype [8,11,14,28]. Interestingly, PPQ-resistant mutations often re-sensitize parasites to CQ, despite the presence of the K76T mutation [8,9]. These parasites also carry mutant Kelch13 (K13) that mediates delayed parasite clearance following treatment with an ART derivative, and frequently harbor multicopy *plasmepsins II* and *III* that appear to augment the degree of PPQ resistance [14,29–31]. The contemporary PPQ-resistant mutations include F145I, which mediates high-level resistance but causes a significant growth defect, as well as T93S and I218F, which mediate lower resistance yet incur a minimal fitness cost [9]. These latter two are rapidly ascending as the dominant PfCRT mutations in SE Asia [4,14], highlighting the continued evolution of the PfCRT haplotype in response to drug pressure and competing growth rates between parasites.

PPQ and CQ are weak-base 4-aminoquinolines that accumulate in the asexual blood stage parasite's acidic digestive vacuole, where they bind to toxic heme species produced due to the endocytosis and degradation of host hemoglobin, thereby preventing heme incorporation into chemically inert hemozoin [15]. PfCRT is a digestive vacuole protein with multiple transmembrane domains and a central negatively-charged cavity comprised of four pairs of antiparallel helices, as determined using cryo-electron microscopy [32]. Most PPQ- or CQ-resistant mutations line this cavity, through which positively-charged drugs and other solutes including hemoglobin-derived peptides can be effluxed out of the digestive vacuole into the parasite cytosol [32–35].

While PPQ resistance was documented clinically in Cambodia in 2014, reports dating back to the 1980s suggest that resistance previously emerged in Yunnan Province, China, when PPQ was used as a monotherapy to treat CQ-resistant malaria [36–38]. This region borders Myanmar, Laos and Vietnam and was the most difficult region to control malaria transmission in China [39]. We hypothesized that mutations in PfCRT may explain the early PPQ resistance in Yunnan Province and that investigating PPQ-resistant isoforms from that region could provide important evolutionary clues when compared to the structure and function of the currently circulating isoforms. Herein, we identify a Chinese PfCRT PPQ-resistant isoform and characterize it using genetic, biochemical, and structural modeling approaches. Inspired by our observations that PPQ-resistant mutations often result in re-sensitization to CQ and reduced parasite fitness, we developed a computational model to conduct evolutionary simulations on empirically determined fitness landscapes. This model allowed us to test the hypothesis that combining PPQ and CQ may be an effective treatment strategy in regions of PPQ resistance by preventing recrudescence of drug-resistant parasites. Collectively, our results provide insights into the pleiotropic role of PfCRT as a multidrug resistance transporter,

inform how drug pressure can alter the within-host evolutionary landscape of PfCRT variants, and suggest that the previously highly-effective, well-tolerated antimalarial CQ could be of use in a combination regimen in SE Asia to mitigate the spread and evolution of drug-resistant infections.

## Results

### A novel PfCRT haplotype from Yunnan Province, China mediates piperaquine resistance

We identified several unique PfCRT isoforms from patient isolates obtained in 2003–2004 in Yunnan Province, China [40]. Using *pfcrt*-specific gene editing [41], we introduced these variant alleles into the well-characterized SE Asian, CQ-resistant Dd2 parasite in place of its endogenous allele (Table 1 and S1 Fig). The Dd2 line used herein expresses 2 copies of *pfdmr1* that encodes the N86Y mutation, as well as a single copy of the *plasmepsin II* gene [8,42]. Isogenic parasite lines were generated that expressed either the PfCRT China E (akin to GB4 + I371R), China B (China E + E75D + A144Y + S220A), or China C (China B + R371I) isoforms. We also generated an experimental Dd2 variant that expresses the A144Y mutation present in China B and C. As control lines we used Dd2$^{Dd2}$, Dd2$^{GB4}$, Dd2$^{Dd2+F145I}$, and Dd2$^{3D7}$, which express the CQ-resistant isoforms Dd2 and GB4, the PPQ-resistant isoform Dd2+F145I, and the wild type (WT) drug-sensitive PfCRT isoform 3D7, respectively [8,17,41]. Of these, GB4 (present in SE Asia and Africa) is the closest to the China isoforms (Table 1).

We tested our isogenic, edited parasite lines for differences in PPQ susceptibility using the piperaquine survival assay (PSA) across a range of PPQ concentrations (1600 nM to 3.1 nM [8,9,12]). Dd2$^{China\ C}$ demonstrated a moderate gain of PPQ resistance (3.4%, 5.9% and 9.8% survival at 800 nM, 200 nM and 100 nM, respectively), compared to the control lines Dd2$^{Dd2}$, Dd2$^{GB4}$, and Dd2$^{3D7}$ that each displayed <1.3% parasite survival across all concentrations (Fig 1A and S1 Table). Dd2$^{China\ E}$ and Dd2$^{China\ B}$ had negligible survival at these concentrations (<1.8% and <1.6%, respectively). The PPQ-resistant control Dd2$^{Dd2+F145I}$ parasite showed significant survival across all concentrations (34.7%, 24.4%, and 18.7% at 800 nM, 200 nM and

**Table 1. Parasite lines used in this study.**

| Parasite Line | Origin | PfCRT amino acid at listed positions | | | | | | | | | | | |
|---|---|---|---|---|---|---|---|---|---|---|---|---|---|
| | | 74 | 75 | 76 | 93 | 144 | 145 | 218 | 220 | 271 | 326 | 356 | 371 |
| Dd2$^{Dd2}$ | SE Asia | I | E | T | T | A | F | I | S | E | S | T | I |
| Dd2$^{Dd2+T93S}$ | Cambodia | I | E | T | S | A | F | I | S | E | S | T | I |
| Dd2$^{Dd2+A144Y}$ | Experimental | I | E | T | T | Y | F | I | S | E | S | T | I |
| Dd2$^{Dd2+F145I}$ | Cambodia | I | E | T | T | A | I | I | S | E | S | T | I |
| Dd2$^{Dd2+I218F}$ | Cambodia | I | E | T | T | A | F | F | S | E | S | T | I |
| Dd2$^{Dd2+S326N}$ | Cambodia | I | E | T | T | A | F | I | S | E | N | T | I |
| Dd2$^{Dd2+T356I}$ | SE Asia/Africa | I | E | T | T | A | F | I | S | E | S | I | I |
| Dd2$^{GB4}$ | SE Asia/Africa | I | E | T | T | A | F | I | S | E | N | I | I |
| Dd2$^{China\ E}$ | China | I | E | T | T | A | F | I | S | E | N | I | R |
| Dd2$^{China\ B}$ | China | I | D | T | T | Y | F | I | A | E | N | I | R |
| Dd2$^{China\ C}$ | China | I | D | T | T | Y | F | I | A | E | N | I | I |
| Dd2$^{3D7}$ | Africa | M | N | K | T | A | F | I | A | Q | N | I | R |

Sequence differences from the CQ-resistant Dd2 and CQ-sensitive 3D7 isoforms are shown in dark and light gray, respectively. The origin of the PfCRT isoform is noted. The T93S, A144Y, F145I, and I218F variants all have individual point mutations added to the Dd2 haplotype. The S326N and T356I variants correspond to the previously reported Cam783 and FCB haplotypes, respectively.

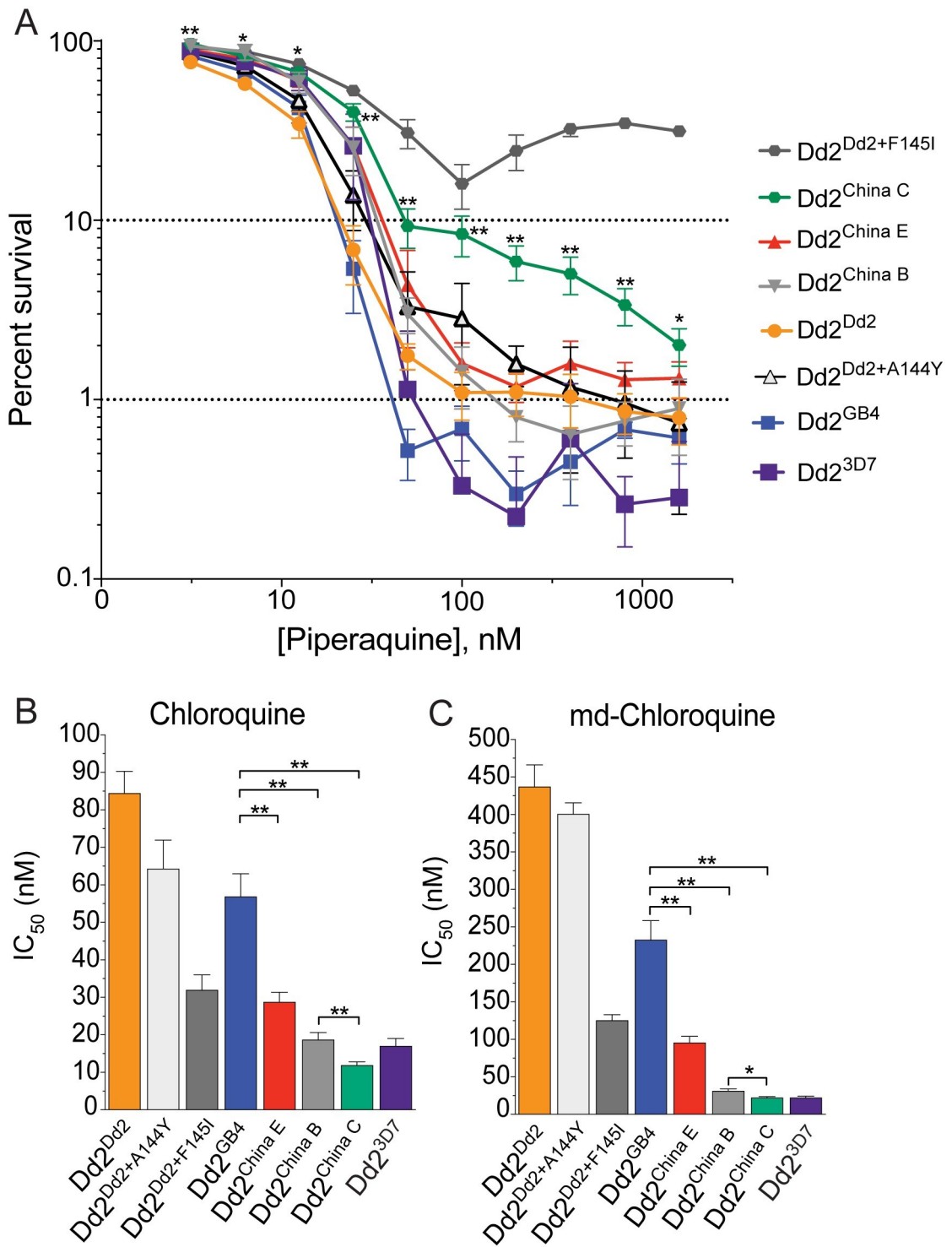

**Fig 1. Dd2^China C parasites demonstrate moderate PPQ resistance and re-sensitization to CQ and md-CQ.** (**A**) Survival of *pfcrt*-modified parasite lines cultured with varying concentrations of PPQ (starting with 0–6 hr rings and treated for 48 hr). The 10% cutoff represents a standard 200 nM threshold for PPQ resistance. Survival values are shown as means ± SEM (S1 Table). *N, n* = 5, 2. (**B, C**) Mean ± SEM IC$_{50}$ values (S2 Table) were calculated from 72-hr dose-response assays for (**B**) CQ and (**C**) md-CQ. *N, n* = 4–7, 2. Statistical significance was determined via two-tailed Mann-Whitney *U* tests as compared to the isogenic line. $^{*}P < 0.05$; $^{**}P < 0.01$.

100 nM, respectively: Fig 1A and S1 Table). The Dd2$^{Dd2+A144Y}$ parasites did not display a statistically significant shift in PPQ susceptibility at any concentration, as compared to the Dd2$^{Dd2}$ control (Fig 1A and S1 Table).

Using 72-hr drug assays with asynchronous parasite lines, we noted significant increases in the PPQ IC$_{50}$ and IC$_{90}$ values of Dd2$^{China C}$, as compared to Dd2$^{Dd2}$ and Dd2$^{GB4}$ (S2 Fig and S2 Table). This suggested that we had uncovered a novel, early PPQ-resistant PfCRT isoform from Yunnan Province, China. This resistance trait is dependent upon the specific combination of PfCRT mutations, as evidenced by the finding that the PPQ-resistant Dd2$^{China C}$ and PPQ-sensitive Dd2$^{China B}$ lines differ only at position 371. Dd2$^{GB4}$, which like Dd2$^{China C}$ has R371I, remained PPQ-sensitive, likely because it lacked the E75D, A144Y and A220S mutations present in Dd2$^{China C}$ (Table 1). Dd2$^{China E}$, which lacks the GB4 R371I mutation and differs from China C at four residues, was also PPQ-sensitive.

Based on the dose-response profile and PSA data, the China C allele did not confer the high-level PPQ resistance we have previously observed with contemporary PfCRT variants. These isoforms, including Dd2+F145I, Dd2+T93S, Dd2+I218F or Dd2+G353V, have been more recently detected in Cambodia and, unlike China C, display biphasic PPQ dose-response curves [8,9].

Of note, the Dd2$^{China C}$ parasites displayed distended digestive vacuoles in both the trophozoite and schizont stages, similar to other PPQ-resistant lines, albeit not as pronounced as those observed in either stage of the highly PPQ-resistant Dd2$^{Dd2+F145I}$ line (S3 Fig) [8,9,43]. The CQ-resistant PPQ-sensitive Dd2$^{Dd2}$ parasites had mildly distended digestive vacuoles in the trophozoite stage but not the schizont stage, as previously observed [43].

## PfCRT variants from Yunnan province have altered susceptibilities to multiple antimalarial drugs

We next tested the susceptibility of these novel isoforms against a panel of antimalarial drugs. Despite having the K76T mutation, all lines expressing the China isoforms were sensitized to CQ and its active metabolite monodesethyl (md)-CQ (Fig 1B and 1C and S2 Table). Notably, Dd2$^{China C}$ was hypersensitized to both agents as compared to Dd2$^{China B}$ and Dd2$^{GB4}$. This is consistent with prior studies showing that the addition of F145I, T93S or I218F to the PfCRT Dd2 isoform resulted in a gain of PPQ resistance and variable levels of re-sensitization to CQ [8,9]. The addition of the A144Y mutation to Dd2 PfCRT did not alter susceptibility to PPQ, CQ or md-CQ (Figs 1 and S2 and S2 Table). No significant differences between the China haplotypes were noted for md-amodiaquine, the active metabolite of the 4-aminoquinoline amodiaquine (S4C Fig and S2 Table). There were also no significant differences between any of these haplotypes or the Dd2$^{Dd2+A144Y}$ strain with respect to mefloquine, dihydroartemisinin, or lumefantrine (S4B, S4D and S4F Fig and S2 Table). Dd2$^{China B}$ and Dd2$^{China C}$ were significantly more sensitive than Dd2$^{China E}$ to quinine (S4A Fig and S2 Table). Notably, Dd2$^{Dd2+A144Y}$ was slightly less susceptible to pyronaridine than Dd2$^{Dd2}$ (S4E Fig and S2 Table). While we have not performed Western blot studies with our *pfcrt*-modified lines, we note that multiple prior studies have assessed *pfcrt*-edited lines that differ in their drug susceptibility profiles and have found no detectable differences in PfCRT protein expression levels using Western blot detection [22,44,45].

## PPQ and CQ uptake profiles are altered in the China C isoform

We next sought to probe these resistance mechanisms at the functional and structural levels with both the China C and the contemporary, highly resistant Dd2+F145I isoforms [32]. These isoforms along with Dd2 and 3D7 were expressed in eukaryotic HEK293 cells and

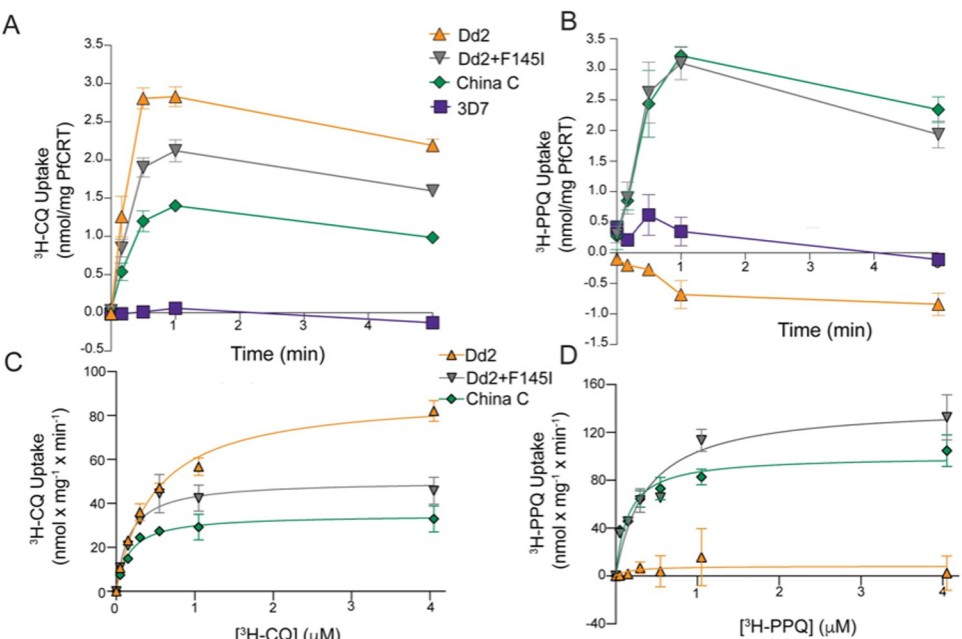

**Fig 2. Functional characterization illustrates distinct CQ and PPQ transport properties for the China C PfCRT protein.** (**A, B**) Time course of (**A**) 93 nM $^3$H-CQ and (**B**) 75 nM $^3$H-PPQ uptake measured with proteoliposomes containing the indicated PfCRT variants. Data are means ± SEM. $N =$ 4. (**C, D**) Kinetic characterization of (**C**) $^3$H-CQ or (**D**) $^3$H-PPQ uptake in proteoliposomes containing the indicated PfCRT isoforms. The initial rate of transport was measured for periods of 10 seconds using $^3$H-CQ or $^3$H-PPQ concentrations ranging from 0.05 to 4.05 μM. Data were fitted to the Michaelis-Menten equation. $K_m$ and $V_{max}$ values are shown as means ± SEM (**S3 Table**). $N =$ 3–4. Transport kinetics of Dd2 or 3D7 PfCRT for PPQ or 3D7 PfCRT for CQ were not determined due to low signal-to-noise ratios and negligible transport.

purified, with Dd2 and 3D7 serving as CQ-resistant and CQ-sensitive controls, respectively (both are PPQ-sensitive) [32]. CQ or PPQ uptake by PfCRT-containing proteoliposomes was used as a surrogate for drug transport from the digestive vacuole to the cytosol [32]. In the proteoliposome system an inwardly directed electrochemical H$^+$ gradient ($\triangle\tilde{\mu}_{H^+}$, consisting of a pH gradient ($\triangle$pH; interior alkaline) and an electrical transmembrane potential ($\triangle\Psi$; negative inside)) were used as driving forces for PfCRT-mediated drug transport, as a means to mimic the physiological conditions under which drugs are transported out of the digestive vacuole (i.e., drug efflux). At 93 nM external $^3$H-CQ, the CQ-resistant PfCRT Dd2 isoform had the highest level of CQ uptake, equating to the highest relative levels of CQ transport (Fig 2A). The CQ-sensitive 3D7 PfCRT isoform demonstrated negligible CQ uptake under these conditions. Dd2+F145I showed intermediate CQ uptake, consistent with the cognate parasite line (Dd2$^{Dd2+F145I}$) having lost most of its CQ-resistant phenotype. One difference was observed with the China C isoform that showed detectable levels of CQ uptake despite the Dd2$^{China\ C}$ line appearing fully CQ sensitive (Figs 1B, 1C and 2A).

The PPQ transport data was more nuanced. The PPQ-sensitive PfCRT isoforms Dd2 and 3D7 both demonstrated negligible transport at 75 nM $^3$H-PPQ, in contrast with the increased PPQ transport observed with the PPQ-resistant Dd2+F145I and China C isoforms (Fig 2B). These findings provide additional evidence that the China C isoform confers PPQ resistance. The finding of similar PPQ transport levels between the Dd2+F145I and China C isoforms was unexpected, given that parasites expressing Dd2+F145I are significantly more PPQ resistant than those expressing China C, as per the PSA data (Fig 1A).

To explore whether the different levels of resistance observed in parasites result from different transport affinities for PPQ, we examined $^3$H-CQ and $^3$H-PPQ transport kinetics with PfCRT-containing proteoliposomes (Fig 2C and 2D). 3D7 PfCRT is not shown as it did not show detectable transport under these conditions [32]. $^3$H-CQ uptake studies showed the highest $V_{max}$ for the Dd2 isoform, followed by Dd2+F145I and then China C, consistent with their relative levels of CQ susceptibility in parasite lines expressing these isoforms (S3 Table). $K_m$ values were higher for Dd2 compared with the other two isoforms (S3 Table). A close association between $V_{max}$ and $IC_{50}$ data was also found for PPQ, where the highest $V_{max}$ values were observed with Dd2+F145I and China C (S3 Table). Transport of 75 nM $^3$H-PPQ by Dd2 was negligible under these conditions, consistent with Dd2$^{Dd2}$ having a PPQ-sensitive phenotype. We were unable to purify the Dd2+A144Y or China B isoforms to perform drug uptake assays with our proteoliposome system. However, we have never been able to detect PPQ uptake with a PPQ-sensitive PfCRT isoform in our assays and thus we would not expect to observe drug transport with either of these isoforms [32].

## The China C PfCRT isoform has distinct conformational changes of the central cavity

We next performed molecular dynamics simulations using the recently solved cryo-EM structure of the 7G8 PfCRT isoform to explore how specific combinations of mutations might alter PfCRT at the structural level [32]. To generate models of Dd2, WT (3D7), China C, and Dd2+F145I, we introduced point mutations into the PfCRT 7G8 structure using the Residue Loop and Mutation tool, followed by a refinement step that consisted of local minimization in implicit solvent with Prime [46]. Simulations suggested that the mutations in China C induced a conformational change in PfCRT, which involved substantial displacement of transmembrane (TM) helix 7, with 144Y (TM3) showing greater atomic distance from I260 (TM7) compared to A144 in Dd2 (Fig 3A). These studies suggest that changes in the cavity shape and size, resulting from the complex interplay of mutations in China C, could affect drug interactions with PfCRT that underly the differential transport profiles observed with CQ and PPQ. Modeling of China C also predicted a significant shift in both TM2 and TM7 compared to the other PPQ-resistant isoform Dd2+F145I, with the former displaying an increased atomic distance of 144Y from S90 (TM2) and I260 (TM7) (Fig 3B). Although these isoforms both confer PPQ resistance, the cavity conformation appeared quite different, which may explain the differences in PPQ transport between China C and Dd2+F145I. Structural modeling also suggested that, while altered cavity conformation is necessary, multiple different cavity conformations can evolve to transport PPQ. We note that China C differs from its PPQ-sensitive counterpart China B by a sole mutation at position 371, located on a loop at the cavity entrance on the digestive vacuole side (Table 1). However, there was no predicted difference in atomic distance from 144Y to the TM9 residues R371 and 371I in China B and China C, respectively (S5A Fig). Possibly, the change from the positively charged arginine residue to the uncharged isoleucine in residue in China C allows for improved entry of the positively charged PPQ molecule into the transporter cavity. Modeling the cavity electrostatic potentials for the Dd2, Dd2+F145I, China C, and China B isoforms predicted no appreciable differences, suggesting that the different degrees of PPQ susceptibility observed with these isoforms is dependent more so on cavity conformational changes rather than on the cavity charge (S5B Fig).

## PPQ resistance comes at a significant fitness cost

To assess the impact of these mutations on parasite asexual blood stage growth rates, we co-cultured each isogenic, GFP$^-$ *pfcrt*-edited parasite line with a Dd2 eGFP$^+$ reporter line [47]

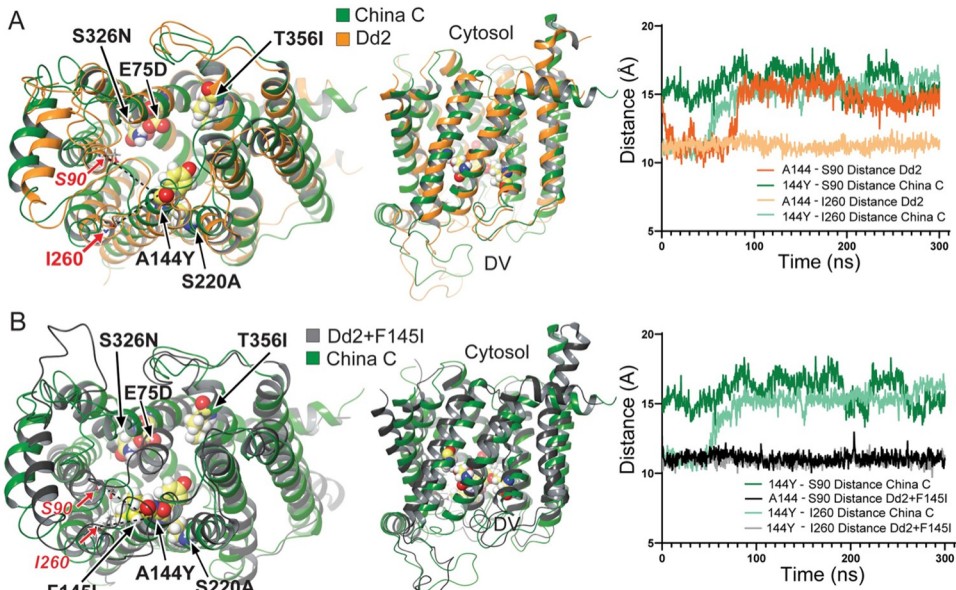

**Fig 3. Molecular dynamics simulations illustrate unique central cavity conformations between the PPQ-resistant China C and Dd2+F145I. PfCRT isoforms.** (**A**) Simulations of molecular dynamics on the 7G8 structure harboring mutations in the China C or Dd2 isoforms and modelled over 300-nanosecond (ns) trajectories, indicating equilibrium positions of protein side chains and predicted distances between position 144 and residues on proximal helices. (**B**) Molecular dynamics simulations for the 7G8 structure harboring the China C or Dd2+F145I mutations, also showing equilibrium positions of protein side chains and predicted distances between position 144 and residues on proximal helices.

(Fig 4A). Cultures were propagated at a ~1:1 starting ratio of GFP⁻ to GFP⁺ parasite lines and sampled every 2 to 3 days for 26 days (13 parasite generations), with the percent of eGFP⁺ parasites determined using flow cytometry. The Dd2$^{3D7}$ line was observed to be the fittest, followed by Dd2$^{Dd2}$, in terms of outcompeting the eGFP⁺ line whose growth was slowed by the expression of the GFP reporter [8]. The Dd2 lines expressing the PfCRT GB4, China E, Dd2 +A144Y and China B isoforms all had comparable fitness profiles and were all more fit than the reporter line. Strikingly, the PPQ-resistant Dd2$^{China\ C}$ line was as unfit as the Dd2$^{Dd2+F145I}$ line that was already known to have a major growth defect [8].

## Empirically determined fitness landscape in the absence of drug

We next created an empirically derived fitness landscape in the absence of drug by leveraging our collection of isogenic PfCRT-edited Dd2 parasite lines to explore how specific PfCRT mutations alter the parasite fitness landscape. For each, we had CQ and PPQ dose-response and proliferation data. Each parasite line was represented as a node and was connected to all known evolutionary neighbors (i.e. those where PfCRT differed by a single amino acid) (Fig 4B). In addition to Dd2$^{Dd2}$, we included the following single amino acid variants: the contemporary, highly PPQ-resistant Dd2$^{Dd2+F145I}$; the less resistant, more fit and increasingly more prevalent Dd2$^{Dd2+T93S}$ and Dd2$^{Dd2+I218F}$; and the GB4 evolutionary intermediates Dd2$^{Dd2+S326N}$ and Dd2$^{Dd2+T356I}$. Dd2$^{Dd2+A144Y}$, which unlike the others has not been described in the field, was included to explore the role of the A144Y mutation present in China B and C. Although Dd2$^{China\ B}$ and Dd2$^{China\ C}$ only differ by one amino acid, they differ by four and three amino acids, respectively, from their nearest neighbor Dd2$^{GB4}$. To connect the landscape, we created two hypothetical intermediates (HI1 and HI2), whose growth rates were linearly

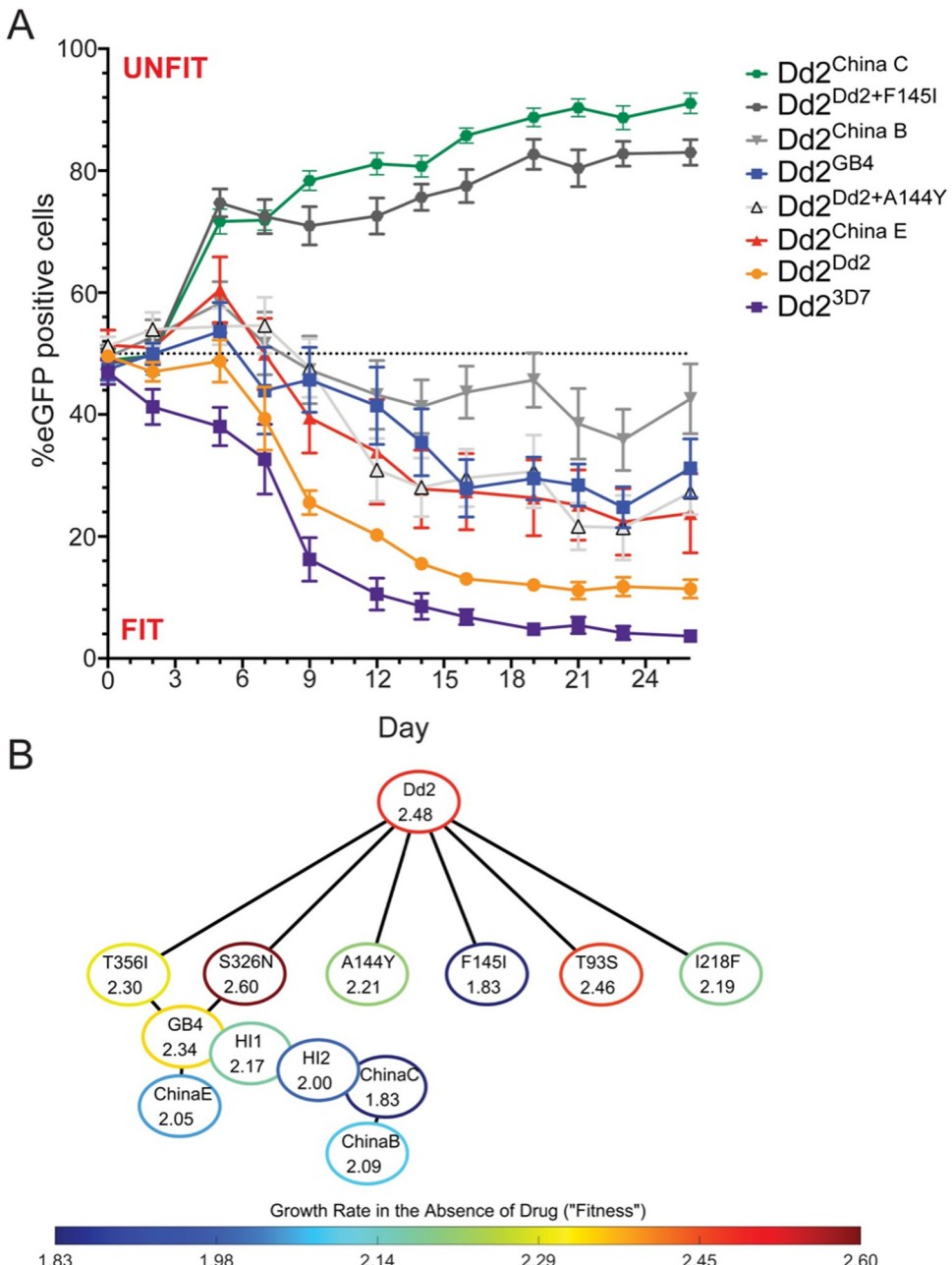

**Fig 4. PfCRT mutations reduce parasite proliferation rates *in vitro*. (A)** Edited *pfcrt* parasites were co-cultured with a GFP-expressing Dd2 line. Cultures were sampled every 2–3 days up to day 26 and were analyzed by flow cytometry. The mean ± SEM percentage of GFP+ parasites was plotted over time. *N, n* = 4, 3. Values above the 50% dashed line indicate a line that was less fit than the GFP+ parasites. Dd2China C and Dd2Dd2+F145I were significantly less fit than the Dd2 GFP reporter line. **(B)** Visualization of the fitness landscape of the parasite lines in the absence of drug. Each node demonstrates the *pfcrt* allele being expressed in an isogenic Dd2 parasite line and its absolute 24-hr growth rate; nodes are outlined in colors that indicate these growth rates. Edges connect nodes that differ by only a single PfCRT amino acid. Two hypothetical intermediate nodes (HI1 and HI2) were linearly interpolated to make the landscape fully connected, by creating a bridge between GB4 and China C, which differ by 3 amino acid substitutions.

interpolated between Dd2GB4 and Dd2China C (Fig 4B). In the absence of drug, parasites expressing the Dd2, Dd2+S326N or Dd2+T93S PfCRT isoforms had the fastest expansion rates, followed by those expressing Dd2+T356I or GB4. Parasites expressing the PfCRT China

C or Dd2+F145I isoforms had the slowest expansion rates (Fig 4B). These growth rates agreed with earlier studies [8,9,17], giving confidence to our calculations.

## Simulations on empirically determined PfCRT fitness landscapes

Our observations that PPQ resistance often leads to CQ re-sensitization, and that PPQ-resistant mutations frequently impact parasite fitness, led us to model these data and theoretically examine: first, how PPQ and CQ drug pressure may have shaped the early emergence of PfCRT-mediated PPQ resistance; second, if we could simulate how drug pressure would impact the within-host evolution of particular PfCRT haplotypes; and third, whether the addition of CQ to a PPQ-containing regimen could be a potential strategy to help restore clinical efficacy in areas of PPQ treatment failures [4]. We modeled treatments with PPQ, CQ, or a combination of the two by developing a new multi-parametric simulator termed DARPS2 (see Materials and Methods). Our simulations began when parasites exited the liver, continued during hypothetical treatment, and terminated with either a fatal outcome (triggered at $1.9 \times 10^{12}$ asexual blood stage parasites) or parasite clearance.

Our computational model operated on the following assumptions: 1) drug concentrations were modeled using a single body compartment; 2) drug levels were assumed to be constant for the three days of drug administration. Drug levels then underwent a multi-exponential decay, with concentrations dropping rapidly due to absorption on the day after the last dose and then more slowly due to drug elimination (S5 Table) [48]; 3) the nine PfCRT variant residues (75, 93, 144, 145, 218, 220, 326, 356, 371) were considered biallelic (i.e. expressing either the WT or variant amino acid at each position) (see Table 1); 4) new, unknown, mutant strains emerging during the simulations were considered inviable; and 5) simulations did not incorporate acquired immunity, which is far lower in SE Asia than in high-transmission settings in Africa [49]. This model also excluded the role of *plasmepsins II* and *III* as potential modulators of PPQ resistance, as prior studies with Dd2 parasites have shown that mutant PfCRT can mediate high-grade resistance with only a single copy of these *plasmepsins* [8]. Given that we were assessing within-host evolution of parasite *pfcrt* alleles, we did not include parasite transmission dynamics in this model. Such data would be necessary to make conclusions on a population evolution basis [50].

To simulate the parasite burden during and after treatment, we created empirical fitness landscapes for each parasite line in the presence of PPQ and/or CQ by combining the drug-free fitness landscape and dose-response data (Figs 4B and S6 and S4 Table; see Materials and Methods). Selected treatment regimens used two separate concentrations of PPQ and CQ. 'Low-dose' treatments corresponded to initial blood concentrations of 200 nM PPQ and/or 125 nM CQ. These concentrations were chosen based upon *in vitro* resistance thresholds that had been demonstrated to correlate with clinical resistance [7,28]. 'High-dose' treatment regimens corresponded to initial concentrations of 400 nM PPQ and/or 250 nM CQ. Each treatment was simulated as a once-daily dose for three consecutive days. Low- and high-dose drug concentrations remain at or below the concentrations observed in patient samples *in vivo* [48,51,52]. However, it should be noted that the high-dose concentrations are close to peak serum concentrations for PPQ (539 ng/mL, 539 nM) [53] but well below those observed for CQ (376 ng/mL, 1.18 uM) [48]. Four scenarios were simulated for each drug regimen: 1) monotherapy with either PPQ or CQ, triggered when the parasites reached a predetermined treatment threshold ($10^{11}$ asexual blood stage parasites); 2) late rescue, where one drug was started at the treatment threshold and a rescue treatment with the other drug began if parasites recrudesced to that threshold; 3) sequential treatment, where one drug was started immediately after completion of the first drug; and 4) simultaneous treatment with PPQ and CQ

started at the treatment threshold. All simulations were run stochastically 100 times. Given that the results for sequential and simultaneous treatment were almost identical, our figures demonstrate only simultaneous treatment outcomes while our tables list results for both scenarios.

## Treatment simulations suggest that Dd2$^{China\ C}$ parasites can survive low-dose PPQ monotherapy

Initial simulations were run with parasites expressing the PfCRT China B or C isoforms to assess how PPQ monotherapy may have impacted their within-host evolution. We also examined the potential benefits of adding CQ to treatment regimens. When low-dose regimen simulations began with Dd2$^{China\ B}$, mutant parasites emerged that encoded the Dd2$^{China\ C}$ variant. In all scenarios, however, all parasites were eliminated after treatment with PPQ, and CQ rescue was not required (Fig 5A-5C and S6 Table). When simulations began with PPQ-resistant Dd2$^{China\ C}$, parasites survived low-dose PPQ monotherapy and the hypothetical patient expired in all runs (Fig 5D and Table 2).

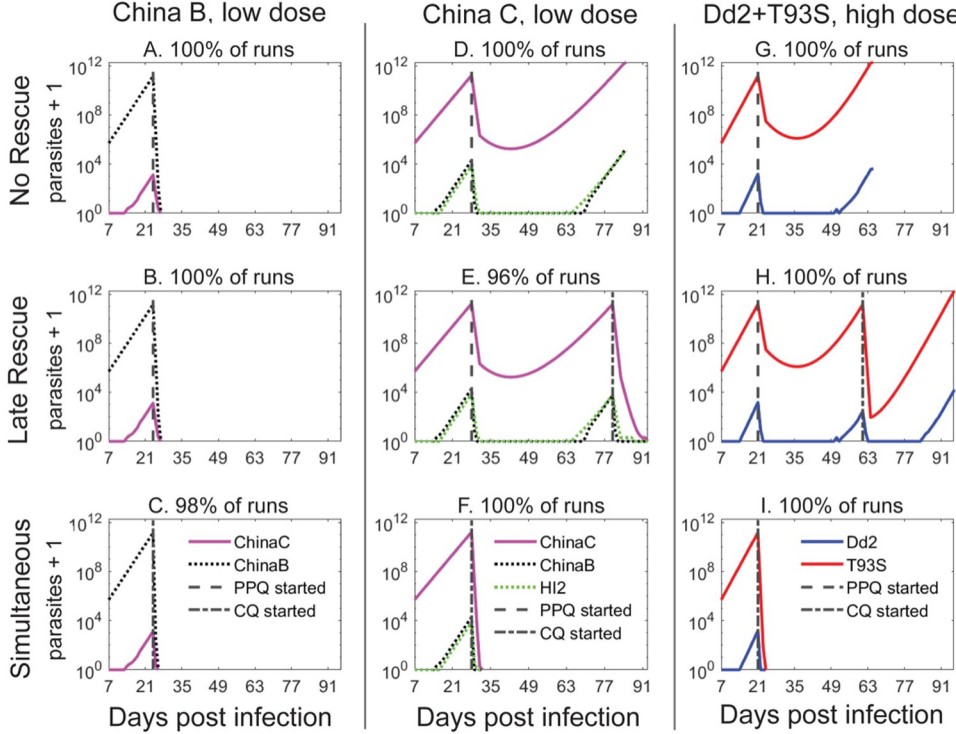

**Fig 5. Evolutionary simulations of empirically determined fitness landscapes with *pfcrt* China alleles or PPQ-resistant Dd2+T93S.** Representative simulation results starting with infection by a monoculture of PPQ-sensitive Dd2$^{China\ B}$ parasites followed by initial treatment with a low-dose regimen of 200 nM PPQ as (**A**) monotherapy; (**B**) plus late rescue with 125 nM CQ; or (**C**) as simultaneous PPQ and CQ combination treatment. Late rescue was not triggered given no parasites survived. Representative simulation results starting with PPQ-resistant Dd2$^{China\ C}$ parasites followed by initial treatment with 200 nM PPQ as (**D**) monotherapy; (**E**) plus late rescue with 125 nM CQ; or (**F**) simultaneous PPQ and CQ combination treatment. Representative simulation results starting with PPQ-resistant Dd2$^{Dd2+T93S}$ parasites followed by initial treatment with a high-dose regimen of 400 nM PPQ as (**G**) monotherapy; (**H**) plus late rescue with 250 nM CQ; or (**I**) simultaneous PPQ and CQ treatment. 100 stochastic simulations were run for each condition, with the percentage frequencies listed for particular outcomes. Dashed vertical lines indicate initiation of each three-day dosing. Results are summarized in Tables 2 and S6.

**Table 2. Summary of simulation outcomes for single strain infections of Dd2$^{Dd2}$, Dd2$^{Dd2+F145I}$, Dd2$^{Dd2+T93S}$, and Dd2$^{China\ C}$.**

| Initial Piperaquine (PPQ) Concentration (nM) | Optional Chloroquine (CQ) Concentration (nM) | Dd2 | Dd2+F145I | Dd2+T93S | China C |
|---|---|---|---|---|---|
| Low PPQ: 200 | No CQ | Dd2+T93S (100%) | Dd2+F145I (100%) | Dd2+T93S (100%) | China C (100%) |
| | Late Rescue CQ: 125 | Dd2+T93S (100%) | Dd2 (89%) | Dd2+T93S (100%) | None (96%) |
| | | | Dd2+F145I (11%) | | HI2 (3%) |
| | | | | | China C (1%) |
| | Sequential CQ: 125 | None (95%) | Dd2+F145I (62%) | Dd2+T93S (100%) | None (100%) |
| | | Dd2+T93S (4%) | None (38%) | | |
| | | Dd2+I218F (1%) | | | |
| | Simultaneous CQ: 125 | None (98%) | Dd2+F145I (62%) | Dd2+T93S (100%) | None (100%) |
| | | Dd2+T93S (1%) | None (38%) | | |
| | | Dd2+I218F (1%) | | | |
| High PPQ: 400 | No CQ | None (66%) | Dd2+F145I (100%) | Dd2+T93S (100%) | None (100%) |
| | | Dd2+F145I (33%) | | | |
| | | Dd2+I218F (1%) | | | |
| | Late Rescue CQ: 250 | None* (66%) | None (97%) | Dd2+T93S (100%) | None* (100%) |
| | | None (3%) | Dd2 (3%) | | |
| | | Dd2 (30%) | | | |
| | | Dd2+I218F (1%) | | | |
| | Sequential CQ: 250 | None (100%) | None (100%) | None (100%) | None (100%) |
| | Simultaneous CQ: 250 | None (100%) | None (100%) | None (100%) | None (100%) |

Simulations were performed with parasites exposed to 8 possible treatment regimens, starting with treatment by piperaquine (PPQ) with optional treatment with chloroquine (CQ). "Late Rescue" was triggered by the simulation when parasites recrudesced to the treatment threshold; "Sequential" means that CQ was dosed immediately after the PPQ dose completed (i.e. on day 4); "Simultaneous" means that PPQ and CQ were administered at the same time. Entries indicate the most dominant strains at the ends of 100 stochastic simulations of each treatment regimen. None indicates that simulated treatment was successful in killing all the parasites. None* means that the scheduled rescue did not occur because all the parasites died before rescue was triggered. HI2, hypothetical intermediate 2.

## Sequential or simultaneous rescue with CQ is an effective option for clearing infections and suppressing the evolution of new resistant parasites

In the late CQ rescue scenario, all Dd2$^{China\ C}$ parasites were cleared in 96% of the low-dose simulations, due to their CQ susceptibility. In the remaining runs, however, resistant parasites evolved and led to fatal outcomes (Fig 5E and Table 2). When the simulated Dd2$^{China\ C}$ infected patient received 3 days of sequential or simultaneous PPQ and CQ treatment, no resistant variants emerged and all parasites were cleared in every run (Fig 5F).

If high-dose regimens (400 nM PPQ ± 250 nM CQ) were used to treat Dd2$^{China\ C}$ infections, then parasites were cleared in all of the runs in all scenarios (monotherapy, late rescue, sequential or simultaneous treatment), likely due to the low level of PPQ resistance, hypersensitization to CQ, and poor parasite growth (Table 2). These initial simulations suggested that sequential or simultaneous treatment with a combination of PPQ and CQ could be effective against PPQ-resistant infections and prevent the emergence of new resistant strains. To explore these observations in the context of the current parasite landscapes in SE Asia, we next

ran simulations starting with the known currently circulating PfCRT variants included in our fitness landscape.

### Treatment simulations suggest that a high-dose combination regimen of PPQ and CQ is effective at clearing current PPQ-sensitive infections and preventing the *de novo* emergence of resistance

In simulations where Dd2$^{Dd2}$ was the initial infecting parasite line, mutations consistently arose as the parasite burden increased. The newly emerged PPQ-resistant variants Dd2$^{Dd2+T93S}$, Dd2$^{Dd2+I218F}$, Dd2$^{Dd2+F145I}$ survived subsequent low-dose PPQ monotherapy treatment and the hypothetical patient expired in all the simulations (Fig 6A). Of note, the relative proportions of each surviving variant at the ends of these simulations mirrored their relative success in areas of PPQ resistance in SE Asia [4,9,14]. This suggested that our model was able to predict how these PfCRT variants could arise in the blood in the context of competing demands of drug resistance and parasite fitness. When CQ rescue was triggered due to parasite recrudescence, the degree of partial CQ sensitization in the Dd2$^{Dd2+T93S}$ line was insufficient to clear all parasites and the patient still expired in all runs (Fig 6B). However, with sequential or simultaneous low-dose

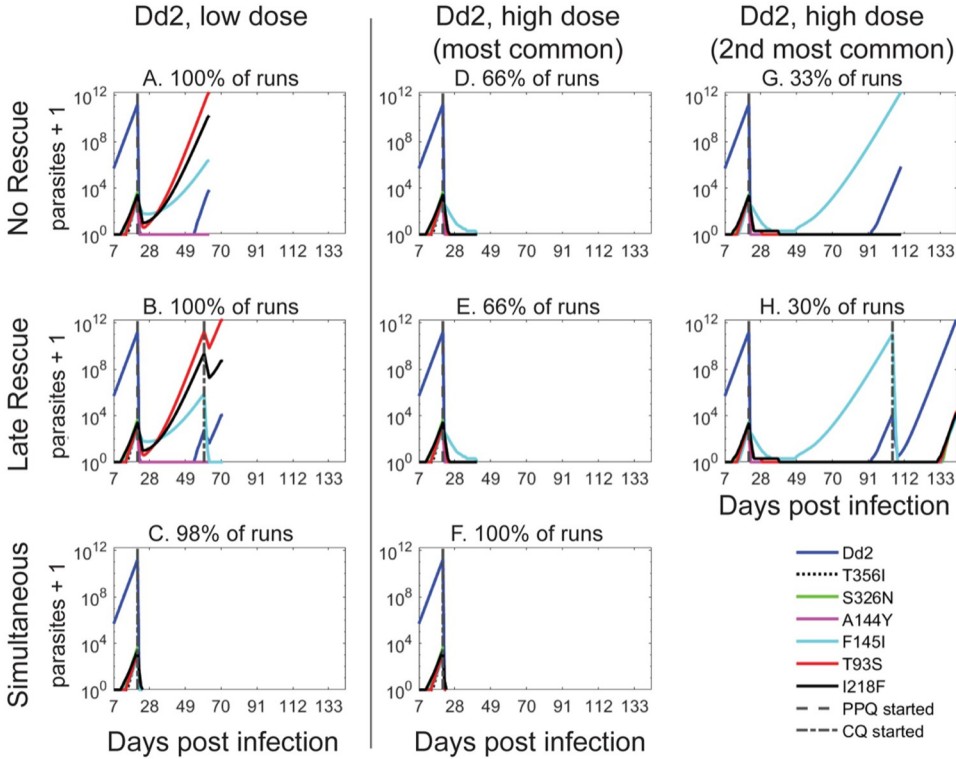

**Fig 6. Evolutionary simulations with Dd2 parasites demonstrate that high-dose PPQ plus CQ can clear infections and prevent the emergence of drug-resistant variants.** Representative simulation results starting with infection by a monoculture of PPQ-sensitive Dd2$^{Dd2}$ parasites followed by initial treatment with a low-dose regimen of 200 nM PPQ as (**A**) monotherapy; (**B**) plus late rescue with 125 nM CQ; or (**C**) simultaneous PPQ and CQ treatment. Representative simulation results starting with Dd2$^{Dd2}$ parasites followed by initial treatment with a high-dose regimen of 400 nM PPQ as (**D, G**) monotherapy; (**E, H**) plus late rescue with 250 nM CQ; or (**F**) simultaneous PPQ and CQ treatment. Panels **D, E, F** show the most common outcome in 100 stochastic simulations of each treatment regimen. Panels **B, D** show the next most common outcome (summarized in Table 2). 100 stochastic simulations were run for each condition, with the percentage frequencies listed for particular outcomes. Dashed vertical lines indicate initiation of each three-day dosing.

PPQ + CQ treatment, the model predicted cure in 98% of runs (Fig 6C), although resistant parasites still led to fatal outcomes in the remaining runs (Table 2).

We next simulated the impact of treating a Dd2$^{Dd2}$ infection with a high-dose regimen. While high-dose PPQ monotherapy cured the hypothetical patient in 66% of the runs, resistant strains emerged in the remainder of runs. In the majority of these runs, the highly PPQ-resistant but less fit Dd2$^{Dd2+F145I}$ parasites emerged and dominated (Fig 6D and Table 2). Even when late CQ rescue was triggered for the runs not cured by the initial high-dose PPQ treatment (Fig 6E), only 3% of simulations resulted in complete parasite clearance (Table 2). In the remaining simulations, Dd2$^{Dd2}$ parasites reemerged and led to a fatal outcome (Fig 6H and Table 2). In striking contrast, no parasites survived sequential or simultaneous high-dose PPQ combined with CQ, leading to complete cure in all runs (Fig 6F and Table 2).

These same four scenarios were examined at both low- and high-dose regimens, starting with the other PPQ-sensitive lines (Dd2$^{Dd2+S326N}$, Dd2$^{Dd2+T356I}$, Dd2$^{Dd2+A144Y}$, Dd2$^{China B}$, Dd2$^{China E}$ and Dd2$^{GB4}$) as the initial infection. In each simulation, all parasites were cleared after PPQ monotherapy, thus CQ rescue was not required (S6 Table). For comparison, we also performed the above simulations with CQ as the initial treatment, optionally followed by late or sequential PPQ rescue. As expected, none of the scenarios using CQ as the initial treatment were successful in always clearing all parasites, except for the high-dose sequential PPQ treatment scenario (S7 and S8 Tables).

## A high-dose combination regimen of PPQ and CQ effectively clears current PPQ-resistant infections and prevents the emergence of mutant PfCRT-mediated resistance

To query whether adding CQ to a PPQ regimen could be a potential strategy in regions with a high prevalence of PPQ resistance, we conducted simulations beginning with PPQ-resistant strains. In Dd2$^{Dd2+T93S}$ infections treated with a low-dose regimen, these parasites persisted in all runs, regardless of any CQ treatment (Table 2). Even in high-dose treatment scenarios, PPQ monotherapy and late CQ rescue were always unsuccessful (Fig 5G-5H). However, when CQ was given sequentially or simultaneously with the initial PPQ treatment, all parasites were eliminated and none ever recrudesced (Fig 5I).

As with Dd2$^{Dd2+T93S}$, the other PPQ-resistant strains Dd2$^{Dd2+F145I}$ and Dd2$^{Dd2+I218F}$ all survived low-dose PPQ treatment without CQ rescue (S7A Fig and Tables 2 and S6). In Dd2$^{Dd2+F145I}$ infections, low-dose PPQ followed by late CQ rescue was always unsuccessful due to either the emerging dominance of "revertant" Dd2$^{Dd2}$ parasites or the persistence of Dd2$^{Dd2+F145I}$ (S7B and S7C Fig). With simultaneous, low-dose CQ and PPQ, all strains cleared in 62% of the simulated runs, with Dd2$^{Dd2+F145I}$ parasites being the most likely to recrudesce in the remaining 38% of runs (S7D and S7E Fig).

In the high-dose regimens, infections with Dd2$^{Dd2+F145I}$ or Dd2$^{Dd2+I218F}$ treated with sequential or simultaneous PPQ+CQ dosing eliminated all parasites with no predicted emergence of new variants (Tables 2 and S6). Taken together, these modeling data provide theoretical evidence that the addition of CQ to a high-dose PPQ regimen might be an effective strategy that is worth further exploration, as the combination was predicted to effectively treat PPQ-resistant infections and suppress the *de novo* within-host evolution of modeled variants.

## Discussion

The evolution of *P. falciparum* resistance to the current first-line antimalarials, including DHA +PPQ in SE Asia, poses a significant threat to malaria treatment and control [4,54]. Herein, we characterize a novel PfCRT isoform (China C) from Yunnan Province, China, that confers

moderate resistance to PPQ, based on elevated PSA survival rates and $IC_{90}$ values, and that could explain the earlier reports of resistance after the use of PPQ monotherapy in this region [37,55]. Biochemical assays and structural modeling illustrated differences in drug transport, as well as electrostatic and conformational changes to the central PfCRT cavity, which mediate different modalities of resistance to PPQ and CQ. Building on the observation that PPQ-resistant parasites are frequently re-sensitized to CQ, we built a simulation model to compare potential combination treatment strategies and assess how differences in drug pressures can shape the evolutionary landscape. Our model predicts that sequential or simultaneous administration of PPQ and CQ for three days would effectively cure infections and suppress the *de novo* emergence of known drug-resistant parasites. Further, these simulation data suggest that DHA+PPQ+CQ could theoretically be an effective triple ACT regimen to force *pfcrt* into an evolutionary trap and prevent the emergence of new PfCRT mutations that would drive high-grade resistance to both PPQ and CQ.

Our studies with PfCRT-reconstituted proteoliposomes reveal that the different rates of CQ transport observed between isoforms mirror the cognate parasite drug susceptibility profiles and may be the major physiological determinant of resistance (Figs 1B, 1C, 2A and 2C). In contrast, PPQ transport data revealed similar levels of 75 nM PPQ transport for both the China C and the Dd2+F145I isoforms. However, the $K_m$ was two-fold higher with Dd2+F145I, consistent with its higher degree of PPQ resistance (Figs 1A and 2B and S3 Table). PfCRT-mediated CQ transport by these China isoforms has also been reported in heterologous yeast and *Xenopus* systems, although to date comparative PPQ transport data were not included [56,57]. Morphologically, Dd2$^{China C}$ and Dd2$^{Dd2+F145I}$ both displayed enlarged digestive vacuoles in the schizont phase, potentially because of a hemoglobin-derived peptide transport defect [34,58,59] (S3 Fig).

Molecular dynamics simulations suggest that PPQ resistance is likely more dependent on conformational changes in the PfCRT cavity than on changes in its electrostatic properties (Figs 3A, 3B and S5). China C demonstrated a drastic shift in TM2 and TM7 as compared to Dd2+F145I, which suggests that multiple cavity conformations can enable PPQ efflux. Changes in cavity shape and size observed between the Dd2 and China C isoforms may also explain why the addition of A144Y present on TM3 and facing the central cavity does not alter PPQ susceptibility in Dd2, given that it lacks the E75D, A220S, S326N and T356I mutations found in China C. One *in vitro* study recently examined the roles of the 326 and 356 residues on PPQ resistance [17]. While the 356 residue was found to have no contribution to PPQ resistance, the S326 residue appeared to provide some protection at lower levels of this drug [17]. This study further supports the hypothesis that PPQ transport is dictated by constellations of residue changes as opposed to being driven by a single point mutation on any given PfCRT isoform. A similar observation was made by Gabryszewski et al. who showed that CQ resistance was only achieved with four or more mutations in PfCRT [21].

China B and C differ only at position 371 located on the digestive vacuole-facing flexible loop. Even though they share the novel A144Y mutation, only China C is PPQ resistant. Their differences in PPQ susceptibility may result either from an allosteric effect of this variant residue on cavity conformation or from the charge difference between the positively charged arginine and the uncharged isoleucine at residue 371, which might impact drug access to the cavity (S5B Fig). The differential impact of these variant residues highlights their complex interplay in conferring drug resistance phenotypes. One caveat to the presented molecular dynamics simulations, however, is that they were performed using the Dd2 isoform for which the structure has not been solved. Dd2 differs from the solved 7G8 structure at positions 72, 326 and 356. Furthermore, there has not yet been a reported drug-bound structure of PfCRT with CQ or PPQ, which would help inform how particular mutations alter drug interactions with the central cavity.

The within-host evolutionary landscape of *P. falciparum* drug resistance represents a balance between competing forces of drug pressure at the population level, differing antimalarial susceptibilities conferred by distinct protein isoforms, and parasite fitness in the absence of drug. On a population level, the evolutionary landscape is also influenced by transmissibility of drug-resistant strains [50]. These factors may help explain why, despite the initial emergence of China C, this low-grade PPQ-resistant, highly unfit isoform did not take hold in the Greater Mekong Subregion despite DHA+PPQ pressure (Figs 1A and 4A). Interestingly, the PPQ-resistant China C isoform was present in only 1 of the 34 isolates identified in the original study [40]. Multiple factors likely contributed to the lack of PPQ-resistant parasites in the study population despite the widespread use of DHA+PPQ. First, during the sample collection period PPQ had been discontinued as a monotherapy ~20 years prior leading to selection of PPQ-sensitive *pfcrt* alleles [55]. Second, despite DHA+PPQ being utilized at that time, ART resistance was still relatively contained, resulting in less selection pressure on PPQ.

Considering the often-dichotomous relationship between PPQ and CQ resistance, we hypothesized that CQ may be an effective addition to antimalarial regimens by serving as an evolutionary trap that could clear infections and prevent the recrudescence of CQ- or PPQ-resistant parasites [60]. This concept has been demonstrated for other antimalarials, including DHODH inhibitors [61], and contributes to the theory behind current triple ACT therapies. These triple ACTs include amodiaquine paired with artemether+lumefantrine or mefloquine paired with DHA+PPQ, which may exert opposing selective pressures [42,62–64]. Of note, the partner drug lumefantrine selects against CQ-resistant PfCRT isoforms, and is associated with the return of CQ sensitivity and wild-type *pfcrt* in Africa [15,62]. Amodiaquine is also a key ACT partner drug, and we postulated that CQ theoretically could be at least as effective, if not more so, given that PfCRT-mediated alterations in sensitization to CQ are more drastic than amodiaquine sensitization (Figs 1B and S4) [8,9,18,19,43].

We tested this hypothesis using our new DARPS2 computational model, based on empirically derived fitness landscapes from isogenic PfCRT-edited Dd2 parasites, to simulate various treatment scenarios with PPQ and CQ. Our simulations suggested that sequential or simultaneous treatment with high-dose drugs, equating to 400 nM PPQ plus 250 nM CQ, was fully effective against PPQ-resistant or -sensitive parasites and prevented treatment failures with drug-resistant PfCRT variants (Tables 2 and S6).

We note that the high-dose PPQ or CQ concentrations used in our model are on the higher end or within the normal range of blood concentrations observed with standard dosing, respectively [51,65]. Individually, CQ and PPQ have been used extensively. Both drugs have been demonstrated to prolong the QT interval, yet neither have led to clinically significant cardiac events [66,67]. It remains to be determined, however, whether the combination of the two medications either dosed simultaneously or sequentially would alter cardiotoxicity. Future studies would also be required to determine tolerability, pharmacokinetics/pharmacodynamics, and safety profiles of the combinations.

Simulations to exploit instances of inverse susceptibility to different drugs, known as evolutionary steering, were earlier developed for drug-resistant bacteria or tumor cells [68,69]. The idea of collateral sensitivity has also been observed in *P. falciparum* parasites treated with proteasome inhibitors targeting distinct catalytic subunits [70]. Our study predicts that exploiting evolutionary steering of *pfcrt* with PPQ and CQ could prevent the emergence of resistant variants. However, with consistent drug pressure new mutations might arise in the PfCRT cavity that could render parasites resistant to both CQ and PPQ.

While the simulation data provide an intriguing framework to assess novel combination treatment regimens, certain limitations of the study should be acknowledged. First, all $IC_{50}$ and modeling data presented herein were based on *in vitro* results from *pfcrt*-edited parasites in the Dd2 background. While this allows an assessment of the *pfcrt* contribution, it does not

take into account how past or current circulating parasite backgrounds can affect drug susceptibility or parasite fitness. Further studies will be needed to assess how combination therapies are influenced by the parasite background, including *plasmepsin* copy number and changes in *pfmdr1* and *pfk13*. It should also be noted that *in vitro* data and simulation models cannot encompass all the complexities that occur within the individual host, including patient co-morbidities, variations in drug metabolism, and intrinsic susceptibility to malaria infection. Notably, our model does not incorporate acquired immune responses that are of particular relevance to high-transmission regions in Africa. However, given the low transmission rates and sparse immunity in SE Asia [49], we would not expect this to substantially alter our simulation outputs for this region. Future studies could use data from these simulations as a starting point for more in-depth *in vivo* studies including in murine infection studies. Our simulations also did not include transmission dynamics, which would be expected to impact the evolution of parasites at the population level. Despite these caveats, our simulations provide a computational tool to explore how parasite density, amino acid mutations, drug pressure, dosing regimens and fitness can combine to shape the evolutionary landscape of parasite populations in the host.

Our open-source computational model is a versatile tool that integrates resistance trajectories, drug pharmacokinetics, parameters of parasite infection, and treatment regimens, to enable the study of how drug concentrations, timing and duration can alter the evolutionary landscape of resistance determinants. Prior studies in the field have modeled the evolution of resistance mutations in *pfdhfr* after exposure to multiple combinations of pyrimethamine or cycloguanil [71–74], or *pfcrt* in response to CQ pressure [21]. Those studies, however, had not modeled how treatment regimens, parasite densities and drug pharmacokinetics affect parasite evolution. Other models have incorporated drug-drug interactions, pharmacokinetics and fixed resistance thresholds to assess how resistance to specific components of a triple ACT might impact treatment efficacy [75], or how resistance might spread in areas of triple ACT use [76]. Those models did not account for parasite fitness that is integral in shaping the parasite resistance landscape. With our model, simulations can be expanded to any current or proposed antimalarials or combinations for which resistance mutations and dose-response and fitness data have been collected in isogenic parasite lines. For instance, this could be expanded to assess treatment regimens with amodiaquine or mefloquine, or to model PfK13 mutants after ART or ozonide treatment [77,78].

Additional refinement of the model, including drug-drug interactions or immunity, can help further explore how the *P. falciparum* resistance landscape evolves under drug pressure. Furthermore, this model allows for prioritization of possible drug combinations prior to testing in lengthy, *in vitro* evolution experiments and even more involved *in vivo* studies. Eventually, it may be possible to design compounds that selectively inhibit the PfCRT transporter and couple them with PPQ or CQ to achieve evolutionary steering, using the model developed herein to screen these combinations *in silico* prior to experimental and clinical validation.

## Materials and methods

### Plasmid construction, parasite culturing and transfections

*pfcrt* was edited using customized zinc-finger nucleases that replace the endogenous allele with a recombinant allele containing the mutations of interest [41]. Dd2 *P. falciparum* parasites were cultured and transfected as described in [79] and S1 Text.

### Piperaquine survival assays and drug susceptibility assays

For PSAs, tightly sorbitol-synchronized ring stage parasites (0–6 hr post invasion) were seeded at 1% parasitemia and 1% hematocrit in 96-well flat-bottom plates containing 2-fold dilutions

of PPQ with a maximum concentration of 1600 nM [8,9,12]. Parasites were incubated for 48 hr at 37˚C, washed, and cultured a further 24 hr. Parasitemias were measured by flow cytometry and percent survival calculated (S1 Text). For drug susceptibility assays, asynchronous, asexual blood stage parasites were plated at 0.3–0.5% parasitemia and 1% hematocrit in 96-well plates, incubated with drug dilutions at 37˚C for 72 hr, and parasitemias measured by flow cytometry (S1 Text).

## PfCRT protein expression and purification

The *pfcrt* 7G8, HB3, Dd2, Dd2+F145I, and China C full-length open-reading frames were cloned into the pEG BacMam vector [80], recombinant P1 baculovirus were prepared as described in [32], and protein purified from HEK293 GnTi-negative cells by $Ni^{2+}$-NTA resin chromatography (S1 Text) [32,81].

## Transport measurements

Purified PfCRT variants were reconstituted in preformed liposomes (composed of *E. coli* total lipids: CHS at a ratio of 94:6 (w/w)) and uptake of $^3$H-CQ (20 Ci/mmol) and $^3$H-PPQ (15 Ci/mmol; both from American Radiolabeled Chemicals, Inc.) was performed as described [32]. Kinetic transport constants ($K_m$ and $V_{max}$) were determined as described in S1 Text.

## Molecular dynamics simulations

Calculations were performed using the Schrödinger molecular modeling suite (version 2019–1) and variant PfCRT protein structures modeled using the Protein Preparation Wizard [82]; S1 Text).

## *In vitro* fitness assays

We measured relative growth rates of *pfcrt*-edited parasite lines using *in vitro* competition assays as described in [8,9] and S1 Text.

## Empirical fitness landscapes

To build empirical fitness landscapes, we used growth and drug assay data generated herein for $Dd2^{Dd2}$, $Dd2^{3D7}$, $Dd2^{Dd2+F145I}$, $Dd2^{Dd2+A144Y}$, $Dd2^{GB4}$, $Dd2^{China\ E}$, $Dd2^{China\ B}$ and $Dd2^{China\ C}$ (Table 1) or previously reported for $Dd2^{Dd2+S326N}$, $Dd2^{Dd2+T356I}$, $Dd2^{Dd2+T93S}$, and $Dd2^{Dd2+I218F}$ [9,17]. For each line, we obtained 72-hr expansion rates under no drug pressure by using the raw data from individual drug assays of asynchronous parasites (S1 Text). To estimate growth rates in the presence of variable concentrations of either CQ or PPQ, we used our available dose-response data (Fig 1) [9,17], normalized data to the absolute growth rate in the absence of drug, and fitted these data using the 5-parameter asymmetric Richards equation (GraphPad Prism 8; S1 Text).

## Simulation model

To simulate parasite growth and evolution in the absence and/or presence of drug, we created a more sophisticated version of DARPS (Discrete Asexually Reproducing Population Simulator) [74], dubbed DARPS2. Each simulation was initiated with a monoculture of one of the 11 parasite lines under investigation, which was intended to simulate a population of parasites that was exiting the liver one week after initial sporozoite infection (here, we used an initial population of $5\times10^5$ new asexual blood stage parasites [83]). During each synchronous discrete timestep, the number of parasites for each given line was updated by multiplying the number

of parasites present in the previous timestep by the net growth rate of the line at the current timestep, where net growth incorporated both reproduction by cell division and drug-mediated cell death. *P. falciparum* has a 48-hr asexual blood stage cycle, and each drug was assumed to be administered once daily. Consequently, we employed a 24-hr timestep and reduced $r_0$ and mutation rates accordingly. In this study, net growth rates were computed at each timestep by multiplying the strain-specific absolute 24-hr growth rates ($r_0$, shown in Fig 4B) by the relative growth rates, computed using the parasite line-specific fitted growth curves at the simulated concentrations of both PPQ and CQ (S4 Table and S6 Fig). Any mutants outside of the studied portion of the fitness landscape (Fig 4B) were considered inviable ($r_0$ = 0.0). In DARPS2, if the number of parasites exceeded a predetermined fatality threshold (assumed herein to be $1.9 \times 10^{12}$ parasites [84,85]), the host was assumed to die and the simulation was terminated; otherwise, simulations were terminated when all parasites had been cleared. When the number of parasites first exceeded a predetermined treatment threshold (here, assumed to be $10^{11}$ [83–85]), treatment was initiated with three days of PPQ and/or CQ. Treatment with three days of the other drug either never occurred ('monotherapy'), was initiated if the number of parasites recrudesced back to the treatment threshold ('late rescue') or was administered as a combination therapy ('simultaneous treatment'). The concentration of each drug in the host was assumed to remain constant at the initial dose level throughout the three days of administration. Subsequently, the drug concentration was assumed to follow a multi-exponential decay, whereby concentrations dropped rapidly due to absorption on day four of treatment, and then more slowly due to elimination. Specifically, we used previously reported median absorption and elimination half-life measurements for PPQ and CQ, determined by fitting a two-compartment model with first-order absorption kinetics to PPQ and CQ data [48], and converted these half-life values from hours to days (S4 Table). During each asexual blood stage cycle, we stochastically simulated single point mutations at biallelic nucleotides in *pfcrt*, using a mutation rate of $2.5 \times 10^{-9}$ per nucleotide per cycle [86]. With this rate, the probability of double mutations within a single codon is negligible. Thus, since we modeled nine PfCRT amino acid substitutions using 24-hr timesteps, we used an effective *pfcrt* mutation rate of $1.125 \times 10^{-8}$ per parasite per timestep (9 potentially mutated amino acids per parasite $\times$ $2.5 \times 10^{-9}$ mutations per nucleotide per cycle $\times$ 0.5 cycles). Open source Matlab code for the DARPS2 simulator is available at [87]. In total, we performed 132,000 simulations (11 initial lines $\times$ 2 orders of drug sequence $\times$ 3 types of rescue regimens $\times$ 2 initial dose levels $\times$ 100 repetitions). Specifically, we simulated two orders of treatment (PPQ or CQ as the initial drug); three types of rescue regimens (no rescue, late rescue, or simultaneous treatment); and two different initial dosing levels (low-dose: PPQ 200 nM and/or CQ 125 nM; or high-dose: PPQ 400 nM and/or CQ 250 nM). We ran 100 stochastic repetitions for each unique combination of experimental settings.

## Supporting information

**S1 Fig. Zinc-Finger Nuclease (ZFN)-mediated editing of *pfcrt*.** (**A**) The *pfcrt* gene was edited using a two-plasmid approach, one containing the donor and the other expressing the *pfcrt*-specific ZFN. Parasites were selected on WR99210 and blasticidin (BSD) and cloned by limiting dilution. (**B**) Three sets of PCRs were performed to confirm editing and the modified locus was verified by Sanger sequencing. Primer sequences are noted in S9 Table.
(PDF)

**S2 Fig. Piperaquine dose-response data.** Mean $\pm$ SEM (**A**) IC$_{50}$ values and (**B**) IC$_{90}$ values were determined by conventional 72-hr dose-response assays performed with asynchronous parasite cultures. *N, n* = 5, 2. PPQ IC$_{90}$ values could not be calculated for the Dd2$^{Dd2+F145I}$

line because of the biphasic nature of its dose-response curve. Statistical significance was determined via two-tailed Mann-Whitney $U$ tests as compared to the isogenic line. $^*P$<0.05, $^{**} P$<0.01. Values are noted in S2 Table.
(PDF)

**S3 Fig. Cell morphology of *pfcrt*-edited parasites.** Dd2[China C] and to a lesser extent Dd2[China B] have distended, translucent digestive vacuoles in both the trophozoite and schizont stages. This is not observed in Dd2[GB4] or Dd2[China E]. Dd2[Dd2+F145I] has the most distended vacuoles in both stages. Of note, the Dd2[Dd2] parasites have mildly distended vacuoles in the trophozoite stage only, as previously reported [43].
(PDF)

**S4 Fig. IC$_{50}$ data for common antimalarials.** Mean ± SEM IC$_{50}$ values (S2 Table) were calculated from 72-hr dose-response assays for: (**A**) quinine; (**B**) mefloquine; (**C**) monodesethyl (md)-amodiaquine; (**D**) dihydroartemisinin; (**E**) pyronaridine; and (**F**) lumefantrine. $N$, $n$ = 4–7, 2. Statistical significance was determined via two-tailed Mann-Whitney $U$ tests as compared to the isogenic line. $^* P$<0.05, $^{**} P$<0.01.
(PDF)

**S5 Fig. Simulations of molecular dynamics and electrostatic potential surfaces of isoform-specific PfCRT cavities.** (**A**) Simulations of molecular dynamics on the 7G8 structure (without the PfCRT-specific Fab antibody fragment) with mutations in the China C and China B isoforms modeled over 300-nanosecond (ns) trajectories, establishing the equilibrium positions of protein side chains and distances between position 144 and position 371. (**B**) Electrostatic surfaces for the solved open-to-digestive-vacuole conformation for the modeled isoforms, predicted at pH 5.0. Images are presented as a vertical slice through the transporter, showing net charges in the cavity and locations of transmembrane (TM) helices. Areas in red represent higher electron density (greater negative charge) while those in blue signify lower electron density. The row below shows the view rotated by 180˚ relative to the top row.
(PDF)

**S6 Fig. Dose response curves for piperaquine (PPQ) and chloroquine (CQ) for the strains included in the study.** Values on the y-axes are normalized relative to the 24-hr growth rates in the absence of drug (r$_0$). Measurements represent the average of measured data points (see S1 Text). Curves were fitted using the 5-parameter asymmetric Richards equation provided in GraphPad Prism 8 software, with lower and upper bounds constrained to 0 and 1, with the exception that Dd2[Dd2+F145I] growth under high PPQ concentrations was fit with a quadratic. The simulator uses the fitted curves (with coefficients shown in S4 Table) to interpolate relative growth rates at any desired concentration (in nM) and then multiplies them by the appropriate r$_0$ to infer absolute 24-hr growth rates for a given strain at a given concentration.
(PDF)

**S7 Fig. Evolutionary simulations based upon empirically determined fitness landscapes with a low-dose regimen of piperaquine (PPQ) with or without chloroquine (CQ) in a host infected with the PPQ-resistant Dd2[Dd2+F145I] strain.** Representative simulation results starting with infection by a monoculture of PPQ-resistant Dd2[Dd2+F145I] parasites followed by initial treatment with a low dose regimen of 200 nM PPQ (**A**) monotherapy; (**B, C**), late CQ (125 nM) rescue, triggered when the parasite burden again reaches the treatment threshold; or (**D, E**), simultaneous PPQ and CQ combination treatment. Panels **B, D** show the most common outcome in 100 stochastic simulations at each treatment regimen. Panels **C, E** show the next most common outcome. 100 stochastic simulations were run for each condition, with

percentage frequencies of a particular outcome shown. Dashed vertical lines indicate when each 3-day dosing began. Strain in this study refers a genetically edited Dd2 parasite with the indicated *pfcrt* allele.
(PDF)

**S1 Table. Piperaquine survival assays values of *pfcrt*-modified parasite strains.**
(PDF)

**S2 Table. Antimalarial IC$_{50}$ and IC$_{90}$ values of *pfcrt*-modified parasite strains.**
(PDF)

**S3 Table. CQ and PPQ transport by PfCRT-containing proteoliposomes.**
(PDF)

**S4 Table. Normalized dose response curves for PPQ and CQ.**
(PDF)

**S5 Table. Drug decay half-lives utilized in the simulator.**
(PDF)

**S6 Table. Summary of simulation outcomes for PPQ treatment of *P. falciparum* strains expressing variant PfCRT isoforms.**
(PDF)

**S7 Table. Summary of simulation outcomes for CQ treatment of *P. falciparum* strains expressing variant PfCRT isoforms.**
(PDF)

**S8 Table. Summary of simulation outcomes for CQ treatment of *P. falciparum* strains expressing variant PfCRT isoforms.**
(PDF)

**S9 Table. List of oligonucleotides used in this study.**
(PDF)

**S1 Text. Supplementary Materials and Methods.**
(PDF)

## Author Contributions

**Conceptualization:** Jennifer L. Small-Saunders, Margaret J. Eppstein, David A. Fidock.

**Data curation:** Jennifer L. Small-Saunders, Ioanna Deni, Jeremie Vendome, Filippo Mancia, Matthias Quick, Margaret J. Eppstein, David A. Fidock.

**Formal analysis:** Jennifer L. Small-Saunders, Laura M. Hagenah, Kathryn J. Wicht, Ioanna Deni, Matthias Quick, Margaret J. Eppstein, David A. Fidock.

**Funding acquisition:** Paul D. Roepe, David A. Fidock.

**Investigation:** Jennifer L. Small-Saunders, Laura M. Hagenah, Kathryn J. Wicht, Satish K. Dhingra, Jonathan Kim, Jeremie Vendome, Eva Gil-Iturbe, Matthias Quick, Margaret J. Eppstein.

**Methodology:** Jennifer L. Small-Saunders, Satish K. Dhingra, Jeremie Vendome, Paul D. Roepe, Margaret J. Eppstein, David A. Fidock.

**Software:** Jeremie Vendome, Margaret J. Eppstein.

**Supervision:** Filippo Mancia, Matthias Quick, David A. Fidock.

**Validation:** Jennifer L. Small-Saunders, Margaret J. Eppstein.

**Writing – original draft:** Jennifer L. Small-Saunders, Laura M. Hagenah, Monica Mehta, Matthias Quick, Margaret J. Eppstein, David A. Fidock.

**Writing – review & editing:** Jennifer L. Small-Saunders, Laura M. Hagenah, Kathryn J. Wicht, Satish K. Dhingra, Jonathan Kim, Paul D. Roepe, Filippo Mancia, Matthias Quick, Margaret J. Eppstein, David A. Fidock.

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
