## [Decision Letter · Decision Letter 0]

15 Nov 2021

Dear Dr. Fidock,

Thank you very much for submitting your manuscript "Evolution of piperaquine-resistant Plasmodium falciparum malaria and modeling strategies to mitigate resistance" for consideration at PLOS Pathogens. As with all papers reviewed by the journal, your manuscript was reviewed by members of the editorial board and by several independent reviewers. In light of the reviews (below this email), we would like to invite the resubmission of a revised version that takes into account the reviewers' comments.

We cannot make any decision about publication until we have seen the revised manuscript and your response to the reviewers' comments. Your revised manuscript is also likely to be sent to reviewers for further evaluation.

Sincerely,

Liwang Cui, PhD

Guest Editor

PLOS Pathogens

Dominique Soldati-Favre

Section Editor

PLOS Pathogens

Kasturi Haldar

Editor-in-Chief

PLOS Pathogens

orcid.org/0000-0001-5065-158X

Michael Malim

Editor-in-Chief

PLOS Pathogens

orcid.org/0000-0002-7699-2064

Reviewer's Responses to Questions

**Part I - Summary**

Reviewer #1: The authors have undertaken a retrospective analysis of PfCRT isoforms identified in parasites isolated from patients in 2003-2004 in Yunnan Province, China. In the original study by Cui et al, 24 isolates were typed and assigned as PfCRT sequences of type China A, 7 as China B, 1 as China C, 1 as China D and 1 as China E.

In summary, the authors generated transfectant parasite lines with a view to determining whether these PfCRT mutations confer decreased PPQ sensitivity and have a fitness cost. They identified China C as showing reduced sensitivity and used biochemical and molecular modelling studies to try to understand the molecular basis for the reduced sensitivity.

Specifically, the authors generated isogenic control and transfectant parasite lines in the Dd2 background. They expressed the PfCRT China E (akin to GB4 + I371R), China B (China E + E75D + A144Y + 138 S220A) and China C (China B + R371I) isoforms.

Dd2_China C demonstrated a moderate gain of PPQ resistance, compared to the control line Dd2_Dd2. It would be useful to indicate the points of significant difference on Figure 1.

Interestingly, Dd2_China B, which differs from Dd2_China B only in an R371I mutation, did not show a statistically significant difference in PPQ sensitivity compared with Dd2_Dd2.

The authors state that “Dd2_China C parasites had distended digestive vacuoles in both the trophozoite and schizont stages, similar to other PPQ-resistant lines, albeit not as pronounced as those in Dd2_Dd2+F145I (S3 Fig).” There does not appear to be a panel for Dd2_Dd2+F145I in S3 Fig. Also, Dd2_Dd2 appears to exhibit a distended digestive vacuole phenotype. Is it mislabelled?

The authors measured PPQ uptake by PfCRT-containing proteoliposomes. The China C PfCRT isoform exhibits a similar level of PPQ uptake as the more highly PPQ resistant control, Dd2+F145I. It would be interesting to also examine PPQ uptake by the China B and Dd2_Dd2+A144Y isoforms as this may provide insight into the reasons these very closely related isoforms do not confer resistance.

The authors modelled the effects of the mutation on the PfCRT structure. They observed differences in the conformation of the cavity, due to the A144Y mutation. The structural modelling analysis fails to give a clear molecular explanation of the sensitivity phenotype. The authors suggest that the A144Y-associated conformational change might underpin PPQ resistance. However, they should comment further on the fact that the A144Y mutation is also present in China B, which is not PPQ resistant. Similarly, the mutation is present in Dd2_Dd2+A144Y (which differs from China C only at T356I) and is not PPQ resistant. What is the predicted effect of the T356I mutation?

Reside 371 is located at the cavity entrance on the digestive vacuole side. The authors should comment whether changes from Arg to Ile might be expected to facilitate approach by the positively changed PPQ. The authors could use their models to examine the surface charge in this region.

Do the authors have information about the levels of expression of PfCRT in the different cell lines?

The authors have developed a computational model, which they have used to simulate the effects of fitness and drug sensitivity differences on parasite burden during and after treatment with combinations of PPQ and CQ. This model generates interesting predictions but may be oversold in terms of its usefulness. For example, the authors’ conclusion that a combination of PPQ and CQ is effective at clearing PPQ-resistant infections is a very optimistic view of the simulations, which showed that resistance is predicted to arise in all scenarios other than the logistically difficult and potentially toxic scenario where patients receive 3 days of simultaneous high-dose PPQ and CQ treatment. In the discussion, the authors note that there is little evidence of cardiotoxicity associated with PPQ treatment. However, it should be noted that it can prolong the QT interval and may increase the toxicity of chloroquine. The conclusions from this section need to be more moderate.

Similarly, the title suggests that the manuscript describes a path for evolution of PPQ-resistant parasite. The currently presented simulated data are not sufficiently strong to support this conclusion.

Reviewer #2: General

The submission by Small-Saunders and colleagues is a detailed exploration of the role of the PfCRT protein in the function of chloroquine (CQ) and piperaquine (PPQ), both related 4-aminoquinoline antimalarials. The evolution of resistance to these drugs is a major issue in the control of malaria caused by Plasmodium falciparum. It has been observed in the field that resistance to these two drugs seems to have a reciprocal relationship, and distinct mutations in the pfcrt gene may be responsible.

A previous paper from many of the same authors (reference 30) described the structure of key parts of the PfCRT protein from some CQ and PPQ resistant parasites, and compared the binding and transport properties of these molecules.

The current submission builds on these technical advances, but changes the focus to explore in detail the properties of a set of parasites that carry versions of PfCRT that differ from one another in single amino acids. The goal is to use a coordinated set of these lab based approaches to illuminate the possible evolutionary history of PPQ resistance. They then use these well-defined parameters that they have measured to consider various ways that the CQ treatment might be used in the field to “steer” evolution from selecting PPQ resistance to an evolutionary dead end instead.

This paper is important. CQ resistance is almost universal in many Pf endemic regions and intensive use of PPQ has been observed repeatedly to favor evolution of CQ susceptibility in those populations. PPQ resistance is especially pervasive in the north-eastern parts of the Greater Mekong Region, but a recent publication has also uncovered declines in PPQ responses in Uganda, as well, so the manuscript addresses an urgent and growing problem in malaria treatment.

Specific

- The manuscript compares a set of 12 highly related parasites with edited genotypes that differ by 1 or more mutations in the pfcrt gene, and/or differences in the canonical PfCRT (SVMNT compared with CVIET residues 72-76).

- The rate of growth in culture without drug(s) and their susceptibility to a range of drugs was measured carefully, and compared to establish these parameters for each cultured line.

- Then, capitalizing on prior work of the group, the biochemical and structural properties of some of the lines are compared directly to try to define differences in binding of CQ and PPQ to the PfCRT proteins and the transport of each drug in a standardized system to estimate transport capacities.

- The solved structures of the PfCRT proteins expressed by the various lines are then modeled and compared to define derangement(s) in the binding pocket of each protein that could allow these differences to explain the CQ and PPQ responses in the cultured lines.

- These data sets are then reported in the figures and tables (and especially within the Supplementary files).

This part of the paper is quite overwhelming, but the authors have provided a number of tables with the genotypes of each cultured parasite line (which I kept visible on my second monitor) and the figures are clear and well defined. The systematic comparisons make it possible to follow the logic that considers the relative importance of particular mutations to differences in growth in the lab with and without drug treatment and so on.

The collection of the data in the manuscript is then entered in a newly created model to compare outcomes of treatments with PPQ and CQ of each of the 12 lines used. The idea is to test the potential success in patients infected with a specific genotype of parasite comparing various combinations and timing of PPQ plus CQ “treatments”. A de novo model was created, and populated with the data trove reported in the first part of the manuscript. The code and the data are both publicly available in the Supplement, and the authors encourage others to use their model and data to try various other parameters.

The goal seems to be to incentivize clinical teams to consider testing combinations of CQ and PPQ as “triple ACTs” with the goal of possibly adding CQ + DHA PPQ to the triple ACTs currently under clinical assessment in both SE Asia and in Africa. That is suggested at the end of the discussion along with other kinds of comparisons that others might want to explore.

The authors point out that fitness of particular parasite genotypes is not (yet ?) considered in the model, since there is no lab based parameter except growth in culture. One genotype of PfCRT (CVIET common in parts of Asia and Africa does seem to be rapidly cleared when not under CQ pressure). But the complexity of parasite growth in humans doesn’t allow any standardized metric, though the authors do mention that immunity to parasites can’t really be studied on a population basis..

They do begin the paper’s background with a Chinese Pf parasite that carries low level resistance to PPQ that apparently evolved under use of PPQ as a monotherapy early in China. Still, I would wish for at least a short paragraph at the end of the discussion that acknowledged the complexity of outcomes in real patients under treatment, especially outside of the very low transmission that is now the case in SE Asia.

That is not a criticism of the current paper- it is a tour de force, to say the least. But, setting expectations for the CQ+PPQ approach in areas of high transmission where many patients carry parasites with different genotypes is more than challenging. Also, they mention that the safety of combining 2 4-aminoquinolines with similar potential toxicity is a worry and has been studied and has not been a real threat to patient safety. However, the levels used in the high concentration version of the model are very high indeed. I was unable to find quickly the maximum allowed exposure to PPQ or CQ. The tolerability of the high concentrations that were most effective in the model might be worth clarifying.

A very small detail- in most of Africa, the predominant parasites are currently CQ sensitive. That is not so much a result of the withdrawal of CQ, but rather that the currently used ACT is artemether- lumefantrine, and that drug has over time selected the CQ sensitive allele repeatedly. Lumefantrine is a somewhat related molecule to CQ and PPQ, so that is worth pointing out as a possible current example of these kinds of evolutionary selection in the real world.

In sum, the paper is excellent and uses a wide range of technical approaches to test a well-defined set of Pf parasites and apply what they learned to the possible application of PPQ and CQ treatments in people. It may stimulate studies in people to test whether similar responses might be possible. That might work in SE Asia where the complexity of human infections is low. The paper makes a reasonable case that it could be worth a try.

Reviewer #3: Small-Sanders et al provide exciting new data on the role of pfcrt variants in mediating resistance of P. falciparum to the antimalarial drug piperaquine. Several improvements in the manuscript are needed for the data to be presented appropriately and clearly.

ABSTRACT

1. Line 32: multi-drug resistant parasites have not "swept across Southeast Asia". In fact,

- the P. falciparum populations in the Mekong countries are crashing; you are describing (in this assertion) a remnant population sliding towards elimination

- these parasite genotypes from Cambodia have not established themselves in Myanmar

- in other SE Asian nations with a reasonable P. falciparum burden piperaquine resistance is not reported (e.g. Indonesia)

- your own data show the Yunnan genotype is of independent origin and has NOT swept in from Cambodia, nor has it swept southwards

- China has now eliminated Pf and Pv malaria, so these Yunnan genotype parasites are presumably extinct (or a residual population may be extant on the Myanmmar side of the border)

Please tone down this unwelcome polemic.

INTRODUCTION

Is it REALLY necessary to introduce yet another three-letter abbreviation (ABS)? This is simply confusing, as it is non-standard to my knowledge. IE (infected erythrocyte) or iRBC are more common, but then in a world of digital publishing why not just write out in full or simply use "parasites" (having defined at outset that this refers to blood-stage asexuals)? Especially since it only occurs a handful of time apart from the Methods section.

2. Line 70-71: CQ was not directly replaced by ACT in most countries, but by SP or SP-AQ or SP-CQ or mefloquine.

3. Line 74: see comment #1.

4. Lines 85-94: I can understand that it is helpful intra-group shorthand to label haplotypes by a corresponding laboratory cell-line ID (GB4, Dd2 etc) but really, for the majority of the field unfamiliar with these lines (or their geographical and temporal origins) it is more precise and informative to use the full pfcrt locus haplotype, As our new understanding of the gene indicates an ever more complex pattern of variation, this becomes even more important. How do you describe a parasite, genotyped directly from a patient, that is "... a bit GB4 with some 7G8 thrown in at the carboxy terminal end"?

The approach I am advocating is less succinct, as precision often is, and requires careful definitiion to start with, but is preferable in my view. This would commprise up to at least 13 amino acid positions if deployed in full (codons 10, 24, 72, 74-76, 93, 145, 218, 220, 271, 326, 371). The wild type would be: QDCMNKFTIAQNR

The imaginary "fully variant" haplotype would be: KYS(I/M)ETISFSESI

The authors may know of other variant positions that could be added to this. Some authors, in providing haplotypes in this form, use bold font and/or underlining to denote variant positions. I hope the authors will consider this approach, much more genetically rigorous, as the new norm in the Fidock lab, which the whole field looks to for establishment of the benchmark in crt studies. Of course, it may be decided to just stick with your Table 1 showing the haplotype for each parasite line or patient-derived genotype mentioned in the paper, and then continuing with the shorthand, but this just kicks the can a very short distance down the lab in my view. The need you hvae found to add positions to your superscript modifiers in Table 1 perfectly illustrates the problem.

5. Line 97: In this example you could write: "These mutations all evolved from the CQ-resistant PfCRT haplotype CVIET at codons 72-76 (refs)."

6. Line 109: briefly define "polytopic" for readers not familiar with trans-membrane protein nomenclature.

RESULTS

7. Line 136: Readers unfamiliar with Dd2 would benefit from a brief description of its pfmdr1 status here (mutations, amplification status).

8. Lines 136-139: this passage would be much easier to read if you just used the full haplotypes, instead of phrase like "China E + E75D + A144Y + 138 S220A"

9. Nice data in Fig. 1. Would be best to have the legend sitting outside the graph boundary (to the right?). It is helpful that the line names are ordered from most-to-least susceptibile, but this only holds for the final concentration.

10. Lines 154-162: results for standard 72h EC50 estimates are interesting and fit well with recent findings of the Cui lab (ref 36). This directly links the F145I allele to the bi-phasic dose-response curves seen in Cambodian studies. Does this render the PSA survival assay unnecessary in parasite populations lacking this parasite phenotype? Are EC50 plus EC90 estimates enough to characterise most parasite isolates? The China E result is fascinating - implying that loss of the aminoquinolone resistant haplotype STI at pfcrt odons 326, 356 and 371 generates a benefit in 72h PPQ dose-response assays compared to the Dd2 haplotype (and in EC50 reduction Fig. 1A). Such parasites are present in Africa (e.g. Sudan and Tanzania in https://pubmed.ncbi.nlm.nih.gov/25253286/) and need investigation perhaps?

11. Lines 164-174: Very interesting and important data. As a range of drugs are used in the study, and more than one assay format, the authors should consider switching from the "IC50" nomenclature to the EC50 form (effective concentration); this is becoming more widely accepted for in vitro suscepibility studies in the antimalarial field. The reasoning is that for most assays it is not possible to ascribe the measured effect to inhibition of growth alone, which would be to ignore variable cytotoxic effects which are drug-dependent in their relative importance.

12. Lines 211-232: This interesting analysis relies on the cryo-EM structural solution for the SVMNT allele (7G8) at codons 72-76. Are the differences from CVIET (Dd2) releavant to understanding the outputs? Are there some caveats due to this?

13. Pages 7-10: These fitness/evolutionary modelling analyses are of interest, particularly the putative interplay between PPQ resistance and CQ susceptibility. However the term "evolutionary" should probably not be used, because the single most important event in the true fitness of a parasite population is transmission to the mosquito. You are really considering blood-stage fitness in isolation of tranmsission potential. As P. falciparum gametocytes are dependent on Hb catabolism for nutrition until at least stage IIb, and have an active digestive vacuole susceptible to CQ and AQ in vitro and in vivo, this needds to be acknowledged and woven into the overall argument. You do not present any data on how transmissible the variants studied are. (This needs acknowledgement, but I am not suggesting this is a flaw in what is presented thus far.)

DISCUSSION

14. Lines 417-418: although triple ACT as explored by Dondorp and colleagues hold some promise, they do not overcome the major issue with our current ACT regimens - three days of artemisinin treatment is insufficient to clear parasites. Would not a better application of your data be the sequential administration of two ACT (DHA-PPQ follwoed by AS-AQ seems the best choice to me) which provides the "evolutionary trap" you require AND gives 6 days of artemisinin. This approach (see https://pubmed.ncbi.nlm.nih.gov/29082016/) has the advantage of using regimens already licensed and available for trials immediately. Given the low numbers of cases in areas blighted by DHA-PPQ resistance, adherence this special 6-day regimen could be feasibly managed in my view. The availability of the AS-AQ regimen already trumps your arguments on the advantages of CQ, as adding the three days of additional artemisinin is in my view crucial - note that it is well known that K13 varaint parasites are fully susceptible in vivo to 7 days artemisinin monotherapy.

15. The authors need a short summary in the Discussion of study weaknesses: use of the Dd2 background (an ancient workhorse), modelling based on a structure from a different allele of pfcrt, evolutionary fitness explored in the absence of a transmission component etc.

**Part II – Major Issues: Key Experiments Required for Acceptance**

Reviewer #1: The insights into the molecular basis for why China C exhibits decreased PPQ sensitivity while closely related isoforms are limited. Analysis of PPQ transport in PfCRT isoforms for China B and Dd2_Dd2+A144Y would strengthen the manuscript.

Reviewer #2: At least some thoughts on how the lab data relate to parasite growth, actual exposure to the two drugs, especially PPQ, and drug responses would make it easier for readers from outside malaria to assess the applicability of the lab data to the field would be very useful, but is not required.

Reviewer #3: None.

**Part III – Minor Issues: Editorial and Data Presentation Modifications**

Reviewer #1: i) The authors refer to a report from 2019 in support of their statement that “More recently, resistance has also emerged to the ACT partner drug piperaquine (PPQ), leading to >50% treatment failures with this combination across the Greater Mekong Subregion”. That study found ~50% treatment failure rates in a couple of areas. The authors should provide additional references to more recent studies or modify their statement.

ii) PSA should be defined at first use.

iii) It would be useful to indicate the points of significant difference between Dd2_China C and Dd2_Dd2 on Figure 1.

iv) The China C isolate (China B + R371I) was identified in only 1 of 33 isolates that were typed in the study led by Cui. The authors should discuss why PPQ R parasites were so rare in Yunan despite DHA-PPQ being widely used.

Reviewer #2: The figures are very busy, to say the least. The figure legends could be improved and that might help, but the many different kinds of lab-based assessments that they used will be a challenge for many readers. I'm not sure that separating the figures into multiple more figures would help

Reviewer #3: See section I.

PLOS authors have the option to publish the peer review history of their article (what does this mean?). If published, this will include your full peer review and any attached files.

Reviewer #1: No

Reviewer #2: No

Reviewer #3: **Yes: **Colin Sutherland
---

## [Editor Report · Decision Letter 1]

13 Jan 2022

Dear Dr. Fidock,

We are pleased to inform you that your manuscript 'Evidence for the early emergence of piperaquine-resistant Plasmodium falciparum malaria and modeling strategies to mitigate resistance' has been provisionally accepted for publication in PLOS Pathogens.

Best regards,

Liwang Cui, PhD

Guest Editor

PLOS Pathogens

Dominique Soldati-Favre

Section Editor

PLOS Pathogens

Kasturi Haldar

Editor-in-Chief

PLOS Pathogens

orcid.org/0000-0001-5065-158X

Michael Malim

Editor-in-Chief

PLOS Pathogens

orcid.org/0000-0002-7699-2064

Minor change: in line 99, change "delayed ART clearance" to "delayed parasite clearance".
---

## [Editor Report · Acceptance letter]

2 Feb 2022

Dear Dr. Fidock,

We are delighted to inform you that your manuscript, "Evidence for the early emergence of piperaquine-resistant Plasmodium falciparum malaria and modeling strategies to mitigate resistance," has been formally accepted for publication in PLOS Pathogens.

Best regards,

Kasturi Haldar

Editor-in-Chief

PLOS Pathogens

orcid.org/0000-0001-5065-158X

Michael Malim

Editor-in-Chief

PLOS Pathogens

orcid.org/0000-0002-7699-2064